# An Incompressible Smoothed Particle Hydrodynamics (ISPH) Model of Direct Laser Interference Patterning

**Cornelius Demuth [1] and Andrés Fabián Lasagni [1,2,\*]**

[1]   Institute of Manufacturing Technology, Technische Universität Dresden, P.O. Box, 01062 Dresden, Germany; cornelius.demuth@iwtt.tu-freiberg.de

[2]   Fraunhofer Institute for Material and Beam Technology IWS, Winterbergstraße 28, 01277 Dresden, Germany

\*   Correspondence: andres_fabian.lasagni@tu-dresden.de; Tel.: +49-351-463-33343

**Abstract:**   Functional surfaces characterised by periodic microstructures are sought in numerous technological applications. Direct laser interference patterning (DLIP) is a technique that allows the fabrication of microscopic periodic features on different materials, e.g., metals. The mechanisms effective during nanosecond pulsed DLIP of metal surfaces are not yet fully understood. In the present investigation, the heat transfer and fluid flow occurring in the metal substrate during the DLIP process are simulated using a smoothed particle hydrodynamics (SPH) methodology. The melt pool convection, driven by surface tension gradients constituting shear stresses according to the Marangoni boundary condition, is solved by an incompressible SPH (ISPH) method. The DLIP simulations reveal a distinct behaviour of the considered substrate materials stainless steel and high-purity aluminium. In particular, the aluminium substrate exhibits a considerably deeper melt pool and remarkable velocity magnitudes of the thermocapillary flow during the patterning process. On the other hand, convection is less pronounced in the processing of stainless steel, whereas the surface temperature is consistently higher. Marangoni convection is therefore a conceivable effective mechanism in the structuring of aluminium at moderate fluences. The different character of the melt pool flow during DLIP of stainless steel and aluminium is confirmed by experimental observations.

**Keywords:**  direct laser interference patterning; nanosecond pulse; metals; process simulation; heat transfer; fluid flow; thermocapillary convection; incompressible smoothed particle hydrodynamics

## 1. Introduction

Microscopic periodic features provide surfaces with superior functionalities. In biological and medical applications, repetitive surface textures improve the biocompatibility of bone implants [1], guide directional cell growth [2] and inhibit bacterial adhesion and biofilm formation [3]. Further advantages of periodic structured surfaces include enhanced light absorption [4], reduced friction [5,6] and anisotropic wetting [7]. These topographies are increasingly considered for the manufacture of functional surfaces, e.g., in biomedical and marine engineering, tribology, optics and aeronautics.

Direct laser interference patterning (DLIP) is a novel method that allows the production of periodic surface structures with feature sizes in the submicron to micron range in a single processing step. For this purpose, the primary beam of a pulsed laser with wavelength $\lambda$ is split into two or more partial beams. The periodic intensity distribution due to the interfering coherent beams is employed to treat a surface

situated in the interference volume. Here, the interference of two beams is considered, resulting in a sinusoidal energy density distribution [8]

$$\Phi(x,y) = 2\Phi_0 \left\{ \cos\left[\frac{4\pi x}{\lambda} \sin\frac{\theta}{2}\right] + 1 \right\}, \tag{1}$$

where $\Phi_0$ is the fluence of each beam and $\theta$ is the intersecting angle between the beams. Accordingly, two-beam interference patterning generates line-like surface structures with a spatial periodicity

$$\Lambda = \frac{\lambda}{2\sin(\theta/2)}, \tag{2}$$

the minimum achievable periodic distance being half of the used wavelength, i.e., $\Lambda = \lambda/2$ for $\theta = \pi$. The fabrication of repetitive microstructures by means of DLIP, using nanosecond pulses of ultraviolet laser radiation, was demonstrated on ceramics [1], polymers [2–4,7], non-metals [5] and metals [6].

A thorough understanding of the process is essential to the precise patterning of surfaces. However, an insight into the physical mechanisms effective during laser processing cannot be gained from experimental observation, especially owing to the short laser pulse duration and the microscopic scale of the surface modification. Nevertheless, numerical simulations enable the detailed investigation of the contemplated physical effects. This approach allows for parameter variations to identify suitable process conditions with regard to texturing a specific substrate, avoiding an excessive consumption of resources in experiments. Grid-based numerical techniques, such as the finite volume and the finite element method (FEM), are usually employed in the simulation of laser material processing [9].

The mathematical model of the DLIP process originally consisted in the heat diffusion equation with the pulsed laser interference irradiation incorporated in the heat source term [8] and sink terms accommodating the latent heat of involved phase changes [10,11]. Using this model, thermal simulations of DLIP were carried out by the FEM to predict the temporal evolution of the temperature distribution near the metal surface and to assess the effect of the laser fluence on the extent of molten and vaporised material regions [10,11]. For aluminium substrates, the significant increase of the absorptivity with temperature necessitates the consideration of a temperature-dependent reflectivity in the model [12]. A detailed description of the FEM simulation of DLIP for metal substrates is presented in [13].

In this work, the thermal modelling outlined so far is expanded for the first time, according to the best of the present authors' knowledge, to comprise the molten bath convection during nanosecond pulsed DLIP of metal surfaces. The additional complexity of this modelling approach is accepted in the prospect of insight into the role of melt pool convection in surface patterning and a profound explanation of the structuring mechanism. Furthermore, the thermofluiddynamic simulations are performed using the mesh-free smoothed particle hydrodynamics (SPH) technique. The application of mesh-free methods, which permit a deformation or even disintegration of the computational domain, is comparatively novel in the simulation of laser processes.

Specifically, the use of SPH in the modelling of laser material interactions is still little explored. The SPH method was originally developed by Gingold, Lucy and Monaghan [14,15] to address astrophysical problems. Notwithstanding its continuing importance in theoretical astrophysics, SPH was subsequently applied to various problems, notably those involving fluid flow, as in turbomachinery [16], coastal and hydraulic engineering [17]. A detailed account of these advances can be found in systematic work [18–21]. Concerning the simulation of laser processing, Chen and Beraun first solved the coupled heat conduction equations governing ultrashort laser pulse action, i.e., of subpico- to picosecond duration, on metal films by the corrective smoothed particle method [22].

The interaction of micro- to millisecond laser pulses or continuous laser irradiation with materials was modelled more frequently. Gross developed an SPH model for the laser cutting of metals [23], which was extended by Muhammad et al. to simulate the micromachining of coronary stents [24]. Considering microsecond pulses as well, Abidou et al. simulated the laser drilling of stainless steel [25]. Earlier on, Tong and Browne used weakly compressible SPH (WCSPH) to model the melt pool flow and heat transfer during laser spot welding [26], i.e., for millisecond pulses. Comparable laser spot welding simulations were performed for metallic workpieces in [27,28]. Regarding continuous irradiation, Yan et al. studied hydrodynamic interactions during laser underwater machining of alumina by SPH [29]. Later on, Hu et al. simulated conduction mode [27,30] and deep penetration laser welding [30] of aluminium. Russell et al. developed a comprehensive SPH methodology for laser based additive manufacturing and applied it to selective laser track melting [28]. Tanaka et al. investigated also heat conduction due to a stationary or moving laser source using the moving particle semi-implicit method similar to SPH [31]. In addition to [28], selective laser melting was modelled by WCSPH in [32–34].

On the contrary, SPH was rarely employed to address the effects of nanosecond laser pulses. The authors of this manuscript previously suggested a thermal model of DLIP for metallic substrates [35]. Cao and Shin predicted the particle motion due to phase explosion during high fluence laser ablation of metals by SPH [36]. Further, Alshaer et al. used SPH to simulate the thermal ablation of aluminium at elevated fluences, particularly the ejection of particles by the recoil pressure [37]. However, the melt pool flow during nanosecond laser irradiation at moderate fluences was not studied to date using SPH. According to the best of the authors' knowledge, an incompressible SPH (ISPH) model was not applied to the laser-induced molten bath flow before, in contrast to weld pool convection [38].

## 2. Mathematical Model

This section presents a mathematical model of the material behaviour during single pulse DLIP. The equations governing the considered physical phenomena are stated and subsequently rewritten in non-dimensional form to reveal the dimensionless numbers characterising the process.

### 2.1. Governing Equations

Throughout the DLIP process, the energy conservation is of fundamental importance. Correspondingly, the mixed enthalpy-temperature formulation of the heat transfer equation reads

$$\rho \frac{\mathrm{d}h}{\mathrm{d}t} = \kappa \Delta T + \dot{q}'''. \tag{3}$$

The conservation of mass and momentum is particularly significant while the substrate is molten due to the effect of the laser pulse. The associated continuity and Navier–Stokes equations are given by

$$\nabla \cdot \boldsymbol{v} = 0, \tag{4}$$

$$\rho \frac{\mathrm{d}\boldsymbol{v}}{\mathrm{d}t} = -\nabla p + \eta \Delta \boldsymbol{v} + \rho \boldsymbol{g}. \tag{5}$$

As a Lagrangian method is employed, the evolution of particle positions is governed by

$$\frac{\mathrm{d}\boldsymbol{x}}{\mathrm{d}t} = \boldsymbol{v}. \tag{6}$$

The substantial derivative $\frac{\mathrm{d}}{\mathrm{d}t} = \frac{\partial}{\partial t} + \boldsymbol{v} \cdot \nabla$ is employed on the left hand side of Equations (3), (5) and (6).

In Equation (3), the specific enthalpy $h$ consists of sensible and latent amounts

$$h = h_{\text{sens}} + h_{\text{lat}} = c_{\text{p}} (T - T_0) + f_{\text{m}} L_{\text{f}} + f_{\text{v}} L_{\text{v}}, \tag{7}$$

where $L_{\text{f}}$ and $L_{\text{v}}$ are the latent heat of fusion and vapourisation, respectively. On the right hand side of Equation (3), $\kappa$ is the thermal conductivity, $T$ the temperature and $\dot{q}'''$ denotes the heat source term. Considering two-beam interference along with a Gaussian temporal shape of the laser pulse and the Beer–Lambert absorption law, the source term is given by

$$\dot{q}''' (x,y,z,t) = \alpha (1 - R) \frac{\Phi (x,y)}{\sigma \sqrt{2\pi}} \exp \left( -\frac{(t - t_{\text{p}})^2}{2\sigma^2} + \alpha (z - z_{\text{surf}}) \right), \tag{8}$$

where $\Phi (x,y)$ is the fluence distribution according to Equation (1), $\alpha$ is the absorption coefficient of the substrate, $R$ is its reflectivity and $\sigma = \tau_{\text{p}} \big/ \left( 2\sqrt{2 \ln 2} \right)$ denotes the standard deviation of the laser pulse with the duration $\tau_{\text{p}}$ (full width at half maximum (FWHM)) at the pulse time $t_{\text{p}}$.

The molten material is conceived as an incompressible fluid, as evident from the mass and momentum conservation in Equations (4) and (5). Moreover, the Oberbeck–Boussinesq approximation is applied, i.e., the density is constant except in the body force term of the momentum Equation (5), where the density varies as a function of temperature according to

$$\rho (T) = \rho_0 \left[ 1 - \beta (T - T_{\text{l}}) \right] \tag{9}$$

with the volumetric thermal expansion coefficient $\beta$. Further, the pressure gradient term in Equation (5) considers the total pressure comprising static and dynamic components, which is represented as

$$p = p_{\text{stat}} + p_{\text{dyn}} = \rho_0 g (z_{\text{surf}} - z) + p_{\text{atm}} + p_{\text{dyn}}, \tag{10}$$

where $p_{\text{atm}}$ is a constant atmospheric reference pressure at the material surface located at $z_{\text{surf}}$.

Employing the wavenumber $k = 2\pi / \lambda$ in Equation (1), inserting Equation (1) in Equation (8), and taking the result and Equations (9) and (10) into account in Equations (3) and (5), respectively, the governing equations take the form

$$\rho_0 \frac{\text{d}h}{\text{d}t} = \kappa \Delta T + (1 - R) \frac{2 \Phi_0 \alpha}{\sigma \sqrt{2\pi}} \left\{ \cos \left[ 2kx \sin \frac{\theta}{2} \right] + 1 \right\} \exp \left( -\frac{(t - t_{\text{p}})^2}{2\sigma^2} + \alpha (z - z_{\text{surf}}) \right), \tag{11}$$

$$\nabla \cdot \boldsymbol{v} = 0,$$

$$\rho_0 \frac{\text{d}\boldsymbol{v}}{\text{d}t} = -\nabla p_{\text{dyn}} + \eta \Delta \boldsymbol{v} - \beta (T - T_{\text{l}}) \rho_0 \boldsymbol{g}, \tag{12}$$

$$\frac{\text{d}\boldsymbol{x}}{\text{d}t} = \boldsymbol{v}.$$

Due to the short time scale of nanosecond laser interference patterning, heat losses due to convection and radiation are neglected and the heat transfer equation is subject to adiabatic boundary conditions

$$- \kappa \frac{\partial T}{\partial \boldsymbol{n}} = 0, \quad \frac{\partial h}{\partial \boldsymbol{n}} = 0. \tag{13}$$

In the presence of melt, homogeneous Neumann boundary conditions are applied to the dynamic pressure and the no-slip condition, i.e., vanishing velocity, is enforced at the bottom of the molten pool

$$\frac{\partial p_\text{dyn}}{\partial \boldsymbol{n}} = 0, \quad \boldsymbol{v} = 0. \tag{14}$$

At the free surface, an inhomogeneous Neumann condition, the Marangoni boundary condition, is employed for the horizontal velocity, cf. [39,40], and a vanishing vertical velocity is prescribed

$$\eta \frac{\partial u}{\partial z} = \frac{\partial \gamma}{\partial x} = \frac{\mathrm{d}\gamma}{\mathrm{d}T} \frac{\partial T}{\partial x}, \quad w = 0. \tag{15}$$

## 2.2. Non-Dimensionalisation

To rewrite the governing equations in dimensionless form, the non-dimensionalisation of the variables position, time, velocity, pressure, temperature and specific enthalpy is performed employing the scales $L$, $L^2/a$, $a/L$, $\rho_0 a^2/L^2$, $T_\text{v} - T_0$ and $c_\text{p,ref}(T_\text{v} - T_0)$, respectively. In particular, the characteristic length scale is specified as the diffusion length $L = 2\sqrt{a\tau_\text{p}}$, where the pulse width $\tau_\text{p}$ (FWHM) is chosen as the laser beam dwell time. Applying the aforementioned scales, the resulting dimensionless variables are

$$x^* = \frac{x}{2\sqrt{a\tau_\text{p}}}, \ t^* = \frac{t}{4\tau_\text{p}}, \ v^* = 2\sqrt{\frac{\tau_\text{p}}{a}}v, \ p_\text{dyn}^* = \frac{4\tau_\text{p}}{\rho_0 a}p_\text{dyn}, \ T^* = \frac{T - T_0}{T_\text{v} - T_0}, \ h^* = \frac{h}{c_\text{p,ref}(T_\text{v} - T_0)}. \tag{16}$$

Using the dimensionless variables in Equation (16), Equations (4), (6), (11) and (12) are obtained in dimensionless form as

$$\frac{\mathrm{d}h^*}{\mathrm{d}t^*} = \Delta T^* + (1 - R)\frac{La}{\sigma^*\sqrt{2\pi}}\left\{\cos\left[\frac{2\pi x^*}{\Lambda^*}\right] + 1\right\}\exp\left(-\frac{\left(t^* - t_\text{p}^*\right)^2}{2\sigma^{*2}} + \alpha^*\left(z^* - z_\text{surf}^*\right)\right), \tag{17}$$

$$\nabla \cdot \boldsymbol{v}^* = 0, \tag{18}$$

$$\frac{\mathrm{d}\boldsymbol{v}^*}{\mathrm{d}t^*} = -\nabla p_\text{dyn}^* + Pr\Delta\boldsymbol{v}^* + PrRa\frac{T^* - T_1^*}{1 - T_1^*}\boldsymbol{e}_z, \tag{19}$$

$$\frac{\mathrm{d}\boldsymbol{x}^*}{\mathrm{d}t^*} = \boldsymbol{v}^* \tag{20}$$

with the dimensionless standard deviation $\sigma^* = \sigma/(4\tau_\text{p})$ of the laser pulse, periodicity $\Lambda^* = \Lambda/L$ and absorption coefficient $\alpha^* = \alpha L$ in Equation (17). In particular, the dimensionless form of the Marangoni boundary condition in Equation (15) is given as [40]

$$\frac{\partial u^*}{\partial z^*} = -\frac{Ma}{1 - T_1^*}\frac{\partial T^*}{\partial x^*}. \tag{21}$$

The dimensionless numbers emerging in the dimensionless Equations (17)–(21), complemented by the Fourier number *Fo* corresponding to the considered physical time, are the Laser number *La*

$$La = \frac{2\Phi_0\alpha}{\rho_\text{ref}c_\text{p,ref}(T_\text{v} - T_0)}, \quad Fo = \frac{t_\text{end}}{4\tau_\text{p}}, \tag{22}$$

the Prandtl number *Pr*, the Rayleigh number *Ra* and the Marangoni number *Ma* defined by

$$Pr = \frac{\nu}{a}, \quad Ra = 8\tau_p \frac{\beta \left(T_v - T_l\right) g \sqrt{a \tau_p}}{\nu}, \quad Ma = -\frac{\mathrm{d}\gamma}{\mathrm{d}T} \frac{T_v - T_l}{\rho_0 \nu} 2\sqrt{\frac{\tau_p}{a}}. \tag{23}$$

In addition, the dimensionless specific enthalpy is obtained as

$$h^* = T^* + f_m Ph_{s/l} + f_v Ph_{l/v}, \tag{24}$$

where the phase change numbers of melting and vapourisation are defined as

$$Ph_{s/l} = \frac{L_f}{c_{p,ref} \left(T_v - T_0\right)}, \quad Ph_{l/v} = \frac{L_v}{c_{p,ref} \left(T_v - T_0\right)}. \tag{25}$$

Finally, the molten and vaporised mass fractions required for Equation (24) and the relation between enthalpy and temperature are given by

$$f_m = \begin{cases} 0 & h^* \leq T_s^* \\ \frac{h^* - T_s^*}{T_l^* - T_s^* + Ph_{s/l}} & T_s^* < h^* < T_l^* + Ph_{s/l} \\ 1 & T_l^* + Ph_{s/l} \leq h^* \end{cases}, \tag{26}$$

$$f_v = \begin{cases} 0 & h^* \leq 1 + Ph_{s/l} \\ \frac{h^* - 1 - Ph_{s/l}}{Ph_{l/v}} & 0 < h^* - 1 - Ph_{s/l} < Ph_{l/v} \\ 1 & 1 + Ph_{s/l} + Ph_{l/v} \leq h^* \end{cases}, \tag{27}$$

$$T^* = \begin{cases} h^* & f_m = f_v = 0 \\ T_s^* + f_m \left(T_l^* - T_s^*\right) & 0 < f_m < 1, f_v = 0 \\ h^* - Ph_{s/l} & f_m = 1, f_v = 0 \\ T_v^* = 1 & f_m = 1, 0 < f_v \leq 1 \end{cases}. \tag{28}$$

## 3. Smoothed Particle Hydrodynamics

The mesh-free SPH method employed in the present work is described in the following. The exposition of the method starts with its fundamentals and then explains the important aspects to be considered to set up a working SPH algorithm. This account comprises the incompressible SPH (ISPH) approach pursued to solve the fluid flow along with a discussion of numerical stability.

### 3.1. Fundamentals of the SPH Method

The mathematical identity underlying the SPH method is the representation of a quantity $\varphi$ by its convolution with the delta distribution $\delta$ given as

$$\varphi\left(x\right) = \int \varphi\left(x'\right) \delta\left(x - x'\right) \mathrm{d}x'. \tag{29}$$

The kernel approximation is performed by replacing the delta distribution inside the integral in Equation (29) with a kernel function $W$, resulting in

$$\varphi\left(x\right) \approx \int_{\Omega_x} \varphi\left(x'\right) W\left(x - x', l_{sm}\right) \mathrm{d}x', \tag{30}$$

where $\Omega_x$ denotes the compact support of $W$ centred at $x$ and defined by the smoothing length $l_{sm}$. The integration in Equation (30) is carried out as a summation over discrete particles representing the computational domain $\Omega$, giving rise to the particle approximation

$$\varphi(x) \approx \sum_{j=1}^{N} \frac{m_j}{\rho_j} \varphi(x_j) W(x - x_j, l_{sm}).$$

(31)

The requirements for the kernel function imposed to establish the approximation in Equation (30) and further desirable properties are [41,42]

$$\lim_{l_{sm} \to 0} W(x - x', l_{sm}) = \delta(x - x') \qquad \text{(approximation of } \delta \text{ distribution)}, \qquad (32)$$

$$W(x - x', l_{sm}) \geq 0 \qquad \forall x' \in \Omega_x \qquad \text{(positivity)}, \qquad (33)$$

$$W(x - x', l_{sm}) = 0 \qquad \forall x' \notin \Omega_x \qquad \text{(compact support)}, \qquad (34)$$

$$W(x - x', l_{sm}) = W(|x - x'|, l_{sm}) \qquad \text{(spherical symmetry)}, \qquad (35)$$

$$W(x - x', l_{sm}) \geq W(x - x'', l_{sm}) \qquad \forall |x - x'| < |x - x''| \qquad \text{(monotonicity)}, \qquad (36)$$

$$\int_{\Omega_x} W(x - x', l_{sm}) \, dx' = 1 \qquad \text{(normalisation)}, \qquad (37)$$

$$W \in C^2 \qquad \text{(smoothness)}. \qquad (38)$$

The existence of a continuous second derivative of the kernel function in condition (38) is imposed [43], and, in particular, is necessary if it is evaluated in second derivative approximations.

In the present work, a quintic B-spline ascribed to Schoenberg [44], introduced in SPH context by Morris et al. [45], is employed. In the notation used by Speith [46], this kernel function consisting of piecewise quintic polynomials is given as

$$W(r, l_{sm}) = \frac{\varsigma}{l_{sm}^d} \begin{cases} (1 - r/l_{sm})^5 - 6\left(\frac{2}{3} - r/l_{sm}\right)^5 + 15\left(\frac{1}{3} - r/l_{sm}\right)^5 & 0 \leq r/l_{sm} < \frac{1}{3}, \\ (1 - r/l_{sm})^5 - 6\left(\frac{2}{3} - r/l_{sm}\right)^5 & \frac{1}{3} \leq r/l_{sm} < \frac{2}{3}, \\ (1 - r/l_{sm})^5 & \frac{2}{3} \leq r/l_{sm} < 1, \\ 0 & 1 \leq r/l_{sm}, \end{cases} \qquad (39)$$

where $r = |x - x'|$, the problem dimension is denoted by $d$ and the normalisation constant results from Equation (37) as [46]

$$\varsigma = \begin{cases} 243/40 & d = 1, \\ 15309/(478\pi) & d = 2, \\ 2187/(40\pi) & d = 3. \end{cases} \qquad (40)$$

Unlike the representation of the physical quantity $\varphi$ itself in Equation (31), the particle approximation of a derivative term involves the first derivative of the kernel function, as elucidated in Section 3.2. In case of the quintic spline kernel function presented above in Equation (39), the first derivative is given as

$$\frac{\partial W(r, l_{sm})}{\partial r} = \frac{-5\varsigma}{l_{sm}^{d+1}} \begin{cases} (1 - r/l_{sm})^4 - 6\left(\frac{2}{3} - r/l_{sm}\right)^4 + 15\left(\frac{1}{3} - r/l_{sm}\right)^4 & 0 \leq r/l_{sm} < \frac{1}{3}, \\ (1 - r/l_{sm})^4 - 6\left(\frac{2}{3} - r/l_{sm}\right)^4 & \frac{1}{3} \leq r/l_{sm} < \frac{2}{3}, \\ (1 - r/l_{sm})^4 & \frac{2}{3} \leq r/l_{sm} < 1, \\ 0 & 1 \leq r/l_{sm}. \end{cases} \qquad (41)$$

*3.2. Approximation of Derivatives*

An approximation of the gradient of a scalar field $\varphi$ can be obtained by inserting $\nabla \varphi$ in Equation (30) and performing an integration by parts [19]

$$
\begin{aligned}
\nabla \varphi \left( \boldsymbol{x} \right) &\approx \int_{\Omega \cap \Omega_x} \nabla_{\boldsymbol{x}'} \left[ \varphi \left( \boldsymbol{x}' \right) \right] W \left( \boldsymbol{x} - \boldsymbol{x}', l_{\mathrm{sm}} \right) \mathrm{d}\boldsymbol{x}' \\
&= \int_{\Omega \cap \Omega_x} \nabla_{\boldsymbol{x}'} \left[ \varphi \left( \boldsymbol{x}' \right) W \left( \boldsymbol{x} - \boldsymbol{x}', l_{\mathrm{sm}} \right) \right] \mathrm{d}\boldsymbol{x}' - \int_{\Omega \cap \Omega_x} \varphi \left( \boldsymbol{x}' \right) \nabla_{\boldsymbol{x}'} W \left( \boldsymbol{x} - \boldsymbol{x}', l_{\mathrm{sm}} \right) \mathrm{d}\boldsymbol{x}' \\
&= \oint_{\partial\Omega \cap \Omega_x} \varphi \left( \boldsymbol{x}' \right) W \left( \boldsymbol{x} - \boldsymbol{x}', l_{\mathrm{sm}} \right) \boldsymbol{n} \left( \boldsymbol{x}' \right) \mathrm{d}\Gamma + \int_{\Omega \cap \Omega_x} \varphi \left( \boldsymbol{x}' \right) \nabla_{\boldsymbol{x}} W \left( \boldsymbol{x} - \boldsymbol{x}', l_{\mathrm{sm}} \right) \mathrm{d}\boldsymbol{x}'.
\end{aligned}
\tag{42}
$$

Concerning the right hand side of Equation (42), the first integral is rewritten as a surface integral according to Gauss' theorem. In addition, the symmetry of the kernel function in Equation (35) implies the property $\nabla_{\boldsymbol{x}'} W \left( \boldsymbol{x} - \boldsymbol{x}', l_{\mathrm{sm}} \right) = -\nabla_{\boldsymbol{x}} W \left( \boldsymbol{x} - \boldsymbol{x}', l_{\mathrm{sm}} \right)$ of the kernel gradient, which is applied to the second integral.

As the surface integral in Equation (42) vanishes if $\boldsymbol{x}$ is far enough from the domain boundary $\partial\Omega$, the straightforward approximation of the gradient of a scalar quantity $\varphi$ results as

$$
\nabla \varphi \left( \boldsymbol{x} \right) \approx \int_{\Omega_x} \varphi \left( \boldsymbol{x}' \right) \nabla_{\boldsymbol{x}} W \left( \boldsymbol{x} - \boldsymbol{x}', l_{\mathrm{sm}} \right) \mathrm{d}\boldsymbol{x}'.
\tag{43}
$$

Consequently, the discrete particle approximation of the gradient of a scalar field $\varphi$ is given as

$$
\nabla \varphi \left( \boldsymbol{x} \right) \approx \sum_{j=1}^{N} \frac{m_j}{\rho_j} \varphi \left( \boldsymbol{x}_j \right) \frac{\boldsymbol{x} - \boldsymbol{x}_j}{\left| \boldsymbol{x} - \boldsymbol{x}_j \right|} \frac{\partial W \left( \boldsymbol{x} - \boldsymbol{x}_j, l_{\mathrm{sm}} \right)}{\partial \left| \boldsymbol{x} - \boldsymbol{x}_j \right|}.
\tag{44}
$$

It is remarkable that the derivative of a quantity, e.g., the gradient of a scalar field $\varphi$ in Equation (44), is evaluated using the quantity at the particle positions and the known derivative of the kernel function, see Equation (41), in the SPH method. However, if the discrete gradient presented in Equation (44) is applied to a constant field, the resulting value is not equal to zero [19].

Intending to solve a system of partial differential equations using a mesh-free particle method, the discrete differential operators are evaluated at the position $\boldsymbol{x}_i$ of a particle of interest $i$ in the following. The zeroth order consistency of the gradient approximation can be recovered by employing the symmetric gradient operator [19,47]

$$
\boldsymbol{G}_i^- \left( \varphi_j \right) = \frac{1}{\rho_i} \sum_{j=1}^{N_i} m_j \left( \varphi_j - \varphi_i \right) \frac{\boldsymbol{x}_i - \boldsymbol{x}_j}{\left| \boldsymbol{x}_i - \boldsymbol{x}_j \right|} \frac{\partial W \left( \boldsymbol{x}_i - \boldsymbol{x}_j, l_{\mathrm{sm}} \right)}{\partial \left| \boldsymbol{x}_i - \boldsymbol{x}_j \right|} \approx \left( \nabla \varphi \right)_i,
\tag{45}
$$

where the shorthand notation $\varphi_i = \varphi \left( \boldsymbol{x}_i \right)$ is used. On the contrary, it is not recommended to approximate the pressure gradient in the momentum Equation (5) by the symmetric gradient operator in Equation (45), as $\boldsymbol{G}_i^-$ does not fulfil the action–reaction principle [47,48], i.e., Newton's third law. Linear momentum conservation can be ensured using the antisymmetric gradient operator [42,48]

$$
\boldsymbol{G}_i^+ \left( \varphi_j \right) = \rho_i \sum_{j=1}^{N_i} m_j \left( \frac{\varphi_i}{\rho_i^2} + \frac{\varphi_j}{\rho_j^2} \right) \frac{\boldsymbol{x}_i - \boldsymbol{x}_j}{\left| \boldsymbol{x}_i - \boldsymbol{x}_j \right|} \frac{\partial W \left( \boldsymbol{x}_i - \boldsymbol{x}_j, l_{\mathrm{sm}} \right)}{\partial \left| \boldsymbol{x}_i - \boldsymbol{x}_j \right|} \approx \left( \nabla \varphi \right)_i,
\tag{46}
$$

which satisfies the action–reaction principle.

In view of the accuracy of the projection-based incompressible SPH approach to be presented in Section 3.4, it is essential that the discrete gradient and divergence operators be skew-adjoint [19,47]. For this reason, the velocity divergence arising in the incompressible SPH scheme is approximated by the symmetric divergence operator [48]

$$
D_i^- \left( v_j \right) = \frac{1}{\rho_i} \sum_{j=1}^{N_i} m_j \frac{\left( v_j - v_i \right) \cdot \left( x_i - x_j \right)}{\left| x_i - x_j \right|} \frac{\partial W \left( x_i - x_j, l_{\text{sm}} \right)}{\partial \left| x_i - x_j \right|} \approx \left( \nabla \cdot v \right)_i,
\tag{47}
$$

where it can be shown that the discrete operators $G_i^+$ and $D_i^-$ are skew-adjoint [19].

Moreover, a second-order differential operator, the Laplacian, is required to solve the governing equations, e.g., for the approximation of the heat conduction term in the heat transfer Equation (3) and the viscous diffusion term in the momentum Equation (5). If a procedure analogous to the one in Equation (42) is followed, the resulting expression involves the second derivative of the kernel function [19,46,47]. Therefore, the approximation is very sensitive to particle disorder [48] and plagued by an undetermined sign of the summands [19,49] as the kernel function exhibits a point of inflexion. On the other hand, a discrete Laplacian could be constructed by composing a gradient and a divergence operator [19,50], i.e., $\Delta_i \approx D_i^- G_j^+$. This exact operator comprises a double summation, which makes it impracticable due to the high computational effort [19,47].

Nevertheless, an approximate Laplacian can be obtained, as first suggested by Morris et al. [45], by combining a finite difference expression for the gradient with an SPH divergence operator [47]. The resulting second-order differential operator employs only the first derivative of the kernel function. This discrete Laplacian originates in the modelling of heat conduction [51,52] and viscous diffusion [45]. Considering the approximation of a general diffusion term, the Laplacian is given by [45,52]

$$
L_i \left( \Gamma_{\varphi,j}, \varphi_j \right) = \sum_{j=1}^{N_i} \frac{m_j}{\rho_j} \left( \Gamma_{\varphi,i} + \Gamma_{\varphi,j} \right) \frac{\varphi_i - \varphi_j}{\left| x_i - x_j \right|} \frac{\partial W \left( x_i - x_j, l_{\text{sm}} \right)}{\partial \left| x_i - x_j \right|} \approx \left\{ \nabla \cdot \left( \Gamma_\varphi \nabla \varphi \right) \right\}_i,
\tag{48}
$$

where $\Gamma_\varphi$ denotes the diffusion coefficient related to the quantity $\varphi$. The Laplacian $L_i$ in Equation (48) represents a simplified form for a spherically symmetric kernel function. Instead of a scalar $\varphi$, the approximate Laplacian can also be applied to a vectorial quantity and the resulting vector is denoted by $L_i$ in this case.

### 3.3. Boundary Conditions

The treatment of boundary conditions in SPH received little attention from its inception in theoretical astrophysics, where boundaries do not play a crucial role [53]. Nevertheless, the application of boundary conditions in an SPH algorithm is a nontrivial task. Particles close to the boundary exhibit an incomplete kernel support due to the truncation by the boundary, the particle deficiency problem [54].

Different approaches were proposed to remedy the incompleteness of the kernel support near the boundaries, including three classical boundary treatment techniques. The first is the generation of mirror particles by direct reflexion of near-wall particles across the boundary, as proposed by Libersky and Petschek [55]. On the other hand, Monaghan [56] introduced boundary particles, which are located at the edges of the computational domain and exert repulsive forces on approaching particles. In addition, Randles and Libersky [54] completed the kernel support of near-wall particles by fixed dummy particles situated beyond the boundaries.

Furthermore, Kulasegaram et al. [57] devised a profound treatment for wall boundaries using a variational formulation of the SPH equations of motion. In this approach, a correction factor, i.e., a function

characterising the completeness of the kernel support, for particles near a wall is introduced and, on its basis, boundary contact forces are evaluated [57]. Considering the distance of a particle from the nearest boundary segment, the correction factor and its derivative are determined using a spline approximation of the original kernel integral [19,57]. Different methods were also developed to account for the case of an intersection of boundary segments [57].

In the present work, boundary conditions to the energy Equation (3) are imposed using fixed dummy particles located beyond the boundaries. Moreover, a combination of edge particles situated right on the boundary and dummy particles is employed to apply the boundary conditions required for the solution of molten pool convection using the incompressible SPH method presented in the following Section 3.4, as suggested by different authors [58,59].

### 3.4. ISPH Approach

The traditional approach of treating incompressible fluid motion using SPH consists of the assumption of a slight compressibility of the fluid and the numerical solution of the compressible conservation equations. In addition, this weakly compressible SPH (WCSPH) method comprises an equation of state to close the system of the compressible continuity and Navier–Stokes equations. However, this procedure gives rise to considerable noise in the pressure field, since small fluctuations in the density field result in large pressure fluctuations due to the stiffness of the equation of state [59]. Remedies suggested for this disadvantage of WCSPH include a particle initialisation algorithm [60] and the introduction of diffusive corrections [61,62] in combination with a particle shifting approach [63].

More recently than WCSPH, the projection method for the numerical solution of the incompressible Navier–Stokes equations developed independently by Chorin [64] and Temam [65] was introduced in SPH context by Cummins and Rudman [50]. The projection method employs the Helmholtz–Hodge decomposition theorem which states that any vector field $w$ on a smoothly bounded domain $\Omega$ can be uniquely decomposed in the form

$$w = u + \nabla p, \tag{49}$$

where $u$ is a solenoidal vector field and parallel to the boundary $\partial\Omega$, and $\nabla p$ is an irrotational vector field [66].

The SPH approach relying on the projection method for solving the equations governing incompressible fluid flow belongs to the incompressible SPH (ISPH) methods. Meanwhile, several ISPH algorithms based on the projection method were proposed using different formulations of the incompressibility constraint, i.e., imposing either a divergence-free velocity field [50] or density invariance [58] or combining both criteria [67]. Other than the projection-based approach, an ISPH method using Lagrange multipliers acting as non-thermodynamic pressures to enforce constant particle volume was presented in [68].

It was recognised that the sole requirement of zero velocity divergence in a projection-based ISPH algorithm leads to substantial particle density variations [67], due to the stretching and compression of particle positions [69]. This particle clustering phenomenon arises due to the ordered motion of the particles along the streamlines [19,69]. By enforcing density invariance in the projection-based ISPH algorithm, the problem can be avoided [67], but this method exhibits a reduced numerical accuracy [69]. Combining both zero velocity divergence and density invariance requirements in an ISPH approach, Hu and Adams could remedy the excessive particle density variations [67].

The latter method being associated with an increased computational effort; Xu et al. proposed an alternative [69]. This approach combines the projection-based ISPH method enforcing a divergence-free velocity field with a slight shift of the particle positions across the streamlines [69]. As this ISPH algorithm is expected to produce acceptable results with reasonable computational effort, it is employed in the

present work. For the numerical solution of the system of governing Equations (4), (6), (11) and (12) in the time step $t^{n+1} = t^n + \Delta t$, the steps of the algorithm are given below. (The gradient of a vector field in Equation (57) is a second-order tensor equal to the Jacobian, i.e., $\nabla v_i^{n+1} = \left[ \nabla v_{1,i}^{n+1} \cdots \nabla v_{d,i}^{n+1} \right]^{\mathsf{T}}$.)

$$\overline{x} = x^n + v^n \Delta t \qquad \qquad \text{position advection} \qquad (50)$$

$$\overline{v} = v^n + (\nu \Delta v^n + \beta \left( T^n - T_1 \right) g e_z) \Delta t \qquad \qquad \text{velocity prediction} \qquad (51)$$

$$\Delta p_{\text{dyn}}^{n+1} = \frac{\rho_0}{\Delta t} \nabla \cdot \overline{v} \qquad \qquad \text{solution of pressure Poisson equation (PPE)} \qquad (52)$$

$$v^{n+1} = \overline{v} - \frac{\Delta t}{\rho_0} \nabla p_{\text{dyn}}^{n+1} \qquad \qquad \text{velocity correction} \qquad (53)$$

$$x^{n+1} = x^n + \left( v^n + v^{n+1} \right) \frac{\Delta t}{2} \qquad \qquad \text{position update} \qquad (54)$$

$$h^{n+1} = h^n + \left( \kappa \Delta T^{n+1} + \dot{q}'''^{,n+1} \right) \frac{\Delta t}{\rho_0} \qquad \qquad \text{enthalpy update} \qquad (55)$$

$$\widetilde{x}_i^{n+1} = x_i^{n+1} + C \alpha R_i \qquad \qquad \text{position shift} \qquad (56)$$

$$\widetilde{v}_i^{n+1} = v_i^{n+1} + \nabla v_i^{n+1} \left( \widetilde{x}_i^{n+1} - x_i^{n+1} \right) \qquad \qquad \text{velocity adjustment} \qquad (57)$$

$$\widetilde{h}_i^{n+1} = h_i^{n+1} + \left( \nabla h_i^{n+1} \right)^{\mathsf{T}} \left( \widetilde{x}_i^{n+1} - x_i^{n+1} \right) \qquad \qquad \text{enthalpy adjustment} \qquad (58)$$

### 3.5. Time Step Criteria

The time step size has to satisfy several constraints in order to ensure the numerical stability of the SPH algorithm. In short, the time step can be determined according to

$$\Delta t = \min \left( 0.25 \frac{l_{\text{sm}}}{v_{\text{max}}}, 0.25 \min_i \sqrt{\frac{l_{\text{sm}}}{|f_i|}}, 0.125 \min_i \frac{l_{\text{sm}}^2}{\nu_i}, 0.125 \min_i \frac{l_{\text{sm}}^2}{a_i}, 0.25 \sqrt{\frac{\rho l_{\text{sm}}^3}{2\pi\gamma}} \right), \qquad (59)$$

the Courant–Friedrichs–Lewy, maximum particle acceleration, viscous and thermal diffusion, and surface tension conditions on the time step being given within brackets.

### 3.6. Neighbour Search

To identify the interacting pairs of neighbouring particles, the cell index method [70] is employed. This approach was described by Hockney and Eastwood [71] with regard to the evaluation of short-range forces in particle methods and first used in the SPH method by Monaghan and Gingold [72]. The principal idea is to subdivide the domain into square cells with a side length equal to or slightly larger than the interaction radius [70,71]. Each particle is assigned to a cell on the basis of its position. The cells are represented by an array of linked lists maintained to keep track of the particles residing in each cell [70–72]. Therefore, the search for all neighbours of a given particle reduces to the examination of the particles in the same cell and in the eight adjacent cells (in 2D) [70,71]. The number of tests required for determining all interacting particle pairs is further halved in the present case of symmetric interactions. The computational effort may be further decreased using an optimal, larger cell size [73] or combining the cell linked lists with a Verlet list, which introduces additional memory requirements [74].

## 4. Numerical Solution of Governing Equations

The intention of the present work is to perform numerical simulations of heat transfer and fluid flow during single pulse DLIP using SPH. A 2D section of the substrate in the $x - z$ plane comprising the

interaction zone due to one period of the interference pattern is considered for this purpose, see Figure 1. This area is discretised using particles as illustrated in Section 4.1. Thereafter, the concrete numerical scheme employed to solve the dimensionless governing Equations (17)–(20) is presented.

The basic strategy to solve the energy Equation (17) in the computational domain throughout the simulation duration is shown in Section 4.2. Shortly after the onset of the nanosecond laser pulse, a thin layer of material adjacent to the surface starts to melt in the vicinity of the interference maximum. In the subdomain representing the molten pool, the solution of the energy equation is a part of a more comprehensive approach to solve the complete set of Equations (17)–(20), which is clarified in Section 4.3.

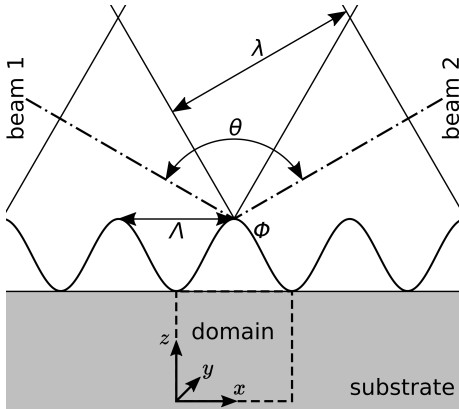

**Figure 1.** Two-beam interference scheme and computational domain.

### *4.1. Discretisation*

As indicated above, the present research considers a rectangular computational domain in the $x - z$ plane, which is discretised using particles. The length of the domain is given by the period $\Lambda$ of the interference pattern in Equation (2), whereas its height amounts to several diffusion lengths $L$, as defined in Section 2.2. To maintain a reasonable computational effort, the employed particle distribution exhibits a local refinement towards the surface in line with the strategy the authors used before in [35]. In particular, the interaction zone, where the substrate is expected to melt due to the action of the laser pulse, is discretised by equidistant fine particles. Despite the associated high computational effort, this idea is followed to allow for equal size particles in the subdomain representing the molten pool.

In detail, the discretisation is performed starting from the bottom of the computational domain, where a row of coarse particles of 1 μm diameter is employed. The discretisation is continued using a successive reduction of the particle size towards the surface, the diameter of the particles in the rows situated above being given by a geometric sequence with a common ratio, or quotient, of $q = 7/9$. A graphical representation of the discretisation is provided in Figure 2, where Figure 2a shows the coarser part referred to so far. As mentioned in the foregoing paragraph, the refinement does not go beyond a minimum particle size, and numerous rows of particles of this size are arranged on a Cartesian lattice to discretise the interaction zone adjacent to the surface, see Figure 2b. The uniform and initially equidistant distribution of fine particles is employed to avoid potential detrimental effects of different particle sizes during the numerical solution of molten pool convection. As a trade-off between an appropriate resolution of the absorption length and a manageable computational effort, the minimum particle diameter is specified as $(7/9)^{22}$ μm ≈ 3.97 nm.

In addition, the discretisation is extended by three rows and columns of dummy particles situated beyond the horizontal and vertical domain boundaries, respectively. Consequently, the kernel function support of the adjacent interior particles is completed by the dummy particles, with the respective particle

diameter being defined by the one of the nearest interior particle. This provision of dummy particles is prescribed by the employed smoothing length. The latter is related to the (dimensionless) kernel support radius $r_W$, which coincides with the number of segments of the radial coordinate in the definition of the spline kernel function in Equation (39).

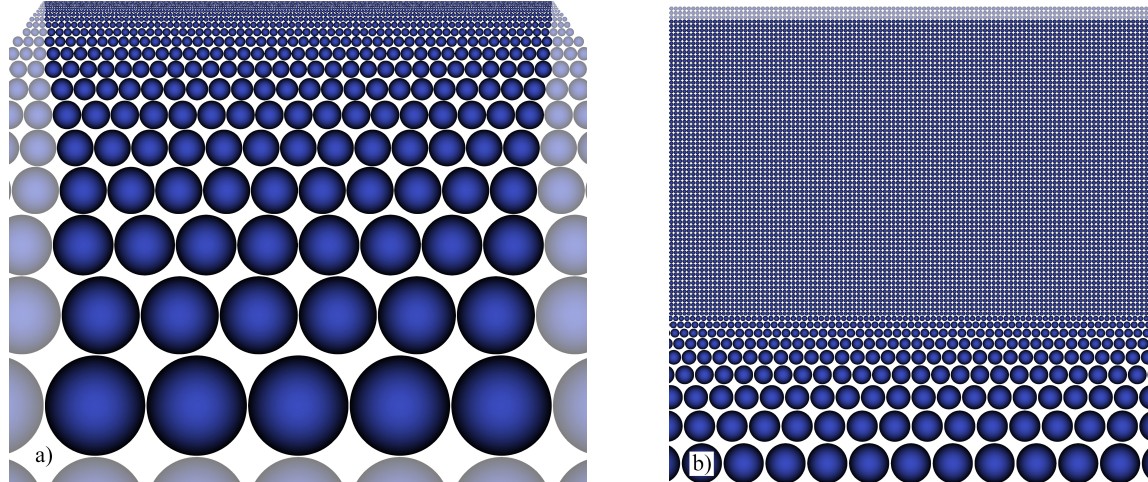

**Figure 2.** Discretisation of computational domain by particles, details of (dummy particles less opaque) (**a**) 5 µm × 4.5 µm, showing coarser particles starting from the bottom, and adjacent dummy particles. (**b**) 440 nm × 440 nm, fine equidistant initial distribution near the surface and coarser particles below.

As stated by Morris, the number of interacting particles should be augmented for a kernel function with larger compact support [75]. For the commonly employed cubic spline kernel function, the kernel support radius is $r_W = 2$ and a typical smoothing length amounts to $l_{\text{sm}}^* = 2.4\Delta x^*$. As the quintic spline kernel function exhibits a larger compact support with $r_W = 3$, an extended smoothing length

$$l_{\text{sm}}^* = 3.75\Delta x^* \tag{60}$$

is used here, where $\Delta x^*$ is the initial separation of particles on a Cartesian grid. The smoothing length specified in Equation (60) corresponds to a neighbourhood of 45 interacting particles in an equidistant rectangular arrangement with particle spacing $\Delta x^*$ in $d = 2$ dimensions.

Note that for interactions between particles of different size, the smoothing length is averaged as explained in the following. Consider two interacting particles $i$ and $j$ separated by $k \in \mathbb{N}\backslash\{0\}$ refinements given by a geometric sequence with common ratio $q < 1$, and assume without loss of generality that the diameter $d_i^*$ of particle $i$ is larger than the diameter $d_j^* = d_i^* q^k$ of particle $j$. The vertical distance between these two particles can be written as

$$z_j^* - z_i^* = k\Delta z_{ji}^{*,\text{av}} = \frac{d_i^*}{2}\left(1 + q\right)\sum_{l=0}^{k-1} q^l = \frac{d_i^*}{2}\left(1 + q\right)\frac{1 - q^k}{1 - q}, \tag{61}$$

i.e., the vertical separation between these particles is the $k$-fold average vertical particle distance

$$\Delta z_{ji}^{*,\text{av}} = \frac{d_i^*}{2k}\left(1 + q\right)\frac{1 - q^k}{1 - q} = \frac{d_j^*}{2k}\left(1 + q^{-1}\right)\frac{1 - q^{-k}}{1 - q^{-1}}. \tag{62}$$

From Equation (60), it is evident that the strength of interaction between particles $i$ and $j$ is non-vanishing only for $k \leq 3$. In particular, the average vertical particle spacing in Equation (62) reduces

to the arithmetic mean of the particle diameters in the case $k = 1$. The above consideration leads to the averaged smoothing length used for the interaction between particles $i$ and $j$

$$l^*_{\text{sm},ij} = \frac{l^*_{\text{sm},i}}{2k}(1+q)\frac{1-q^k}{1-q} = \frac{l^*_{\text{sm},i}}{2k}(1+q)\sum_{l=0}^{k-1}q^l = l^*_{\text{sm},ji}, \tag{63}$$

where the smoothing length $l^*_{\text{sm},i}$ for particle $i$ is given by Equation (60) with the particle separation being equal to the particle diameter, i.e., $(\Delta x^*)_i = d^*_i$.

*4.2. Thermal model*

The energy Equation (17) is solved using the methodology presented earlier by the authors and their co-authors [35]. In particular, the energy Equation (17) in mixed enthalpy–temperature formulation is implicitly integrated in time. Consequently, the discretisation of the dimensionless energy Equation (17) in time for an interior or top edge particle $i$ is written as

$$\frac{h^{*,n+1}_i - h^{*,n}_i}{\Delta t^*} = L^*_i\left(1, T^{*,n+1}_j\right) + \dot{Q}^{*,n+1}_i, \tag{64}$$

where the approximate Laplacian from Equation (48) and the power of the laser heat source per unit mass $\dot{Q}^{*,n+1}_i$ are employed in dimensionless form.

In the discrete Laplacian $L^*_i(1, T^{*,n+1}_j)$, the temperature $T^{*,n+1}_i$ inside the summation is rewritten as a function of the specific enthalpy $h^{*,n+1}_i$ according to Equation (24). Subsequently, Equation (64) can be rearranged to result in an expression for the specific enthalpy at the new time step [35]

$$h^{*,n+1}_i = \frac{h^{*,n}_i + \Delta t^*\left(\dot{Q}^{*,n+1}_i - 2\sum_{j=1}^{N_i}\frac{m^*_j}{\rho^*_j}\frac{T^{*,n+1}_j + f^{n+1}_{\text{m},i}Ph_{\text{s/l}} + f^{n+1}_{\text{v},i}Ph_{\text{l/v}}}{r^{*,n+1}_{ij}}\frac{\partial W\left(r^{*,n+1}_{ij}, l^*_{\text{sm},ij}\right)}{\partial r^{*,n+1}_{ij}}\right)}{1 - 2\Delta t^*\sum_{j=1}^{N_i}\frac{m^*_j}{\rho^*_j}\frac{1}{r^{*,n+1}_{ij}}\frac{\partial W\left(r^{*,n+1}_{ij}, l^*_{\text{sm},ij}\right)}{\partial r^{*,n+1}_{ij}}}. \tag{65}$$

Replacing also $T^{*,n+1}_j$ in Equation (65) with the equivalent terms from Equation (24), a linear system of equations is obtained for the enthalpy field at the new time step, which is constituted by the individual equations for all interior and top edge particles $i$ given as [35]

$$\left(1 - 2\Delta t^*\sum_{j=1}^{N_i}\frac{m^*_j}{\rho^*_j}\frac{1}{r^{*,n+1}_{ij}}\frac{\partial W\left(r^{*,n+1}_{ij}, l^*_{\text{sm},ij}\right)}{\partial r^{*,n+1}_{ij}}\right)h^{*,n+1}_i + 2\Delta t^*\sum_{j=1}^{N_i}\frac{m^*_j}{\rho^*_j}\frac{h^{*,n+1}_j}{r^{*,n+1}_{ij}}\frac{\partial W\left(r^{*,n+1}_{ij}, l^*_{\text{sm},ij}\right)}{\partial r^{*,n+1}_{ij}}$$

$$+ 2\Delta t^*\sum_{j=1}^{N_i}\frac{m^*_j}{\rho^*_j}\frac{\left(f^{n+1}_{\text{m},i} - f^{n+1}_{\text{m},j}\right)Ph_{\text{s/l}} + \left(f^{n+1}_{\text{v},i} - f^{n+1}_{\text{v},j}\right)Ph_{\text{l/v}}}{r^{*,n+1}_{ij}}\frac{\partial W\left(r^{*,n+1}_{ij}, l^*_{\text{sm},ij}\right)}{\partial r^{*,n+1}_{ij}} \tag{66}$$

$$= h^{*,n}_i + \Delta t^*(1-R)\frac{La}{\sigma^*\sqrt{2\pi}}\left\{\cos\left[\frac{2\pi x^{*,n+1}_i}{\Lambda^*}\right] + 1\right\}e^{-\frac{\left(t^{*,n+1} - t^*_{\text{p}}\right)^2}{2\sigma^{*2}} + \alpha^*\left(z^{*,n+1}_i - z^*_{\text{surf}}\right)}.$$

Concerning the iterative solution of this linear system of equations, a few aspects of interest are given in the following. To begin with, the third term on the left hand side of Equation (66) is considered not to contribute to the system matrix, i.e., the molten and vaporised mass fractions are not conceived as a function of the unknown enthalpy in the present iteration. Instead, this term is calculated based on the determination of the molten and vaporised mass fractions from the previous enthalpy iterate.

The formulation in Equation (66) gives rise to a linear system of equations characterised in the following. The system matrix is symmetric, given particles of equal volume (or equal mass in a uniform density approach), its main diagonal elements are positive and the matrix is strictly diagonally dominant. These three properties imply that the matrix is positive definite [76]. The conjugate gradient (CG) method introduced by Hestenes and Stiefel [77] is commonly used to iteratively solve a linear system with a symmetric and positive definite matrix.

In the present work, a preconditioned variant of the CG algorithm [78] in conjunction with a Jacobi preconditioner is employed for the iterative solution of the linear system arising from the discretised dimensionless heat transfer Equation (64). However, only a single CG step is performed, then the quantities in Equations (26)–(28) are updated according to the new enthalpy iterate and this iterative procedure is restarted.

### 4.3. Thermofluiddynamic Model

As indicated in the introductory paragraph of this section, the subsection at hand provides a detailed account of the ISPH scheme employed to solve the system of dimensionless Equations (17)–(20) in a manner analogous to the algorithm given in Equations (50)–(58) for the numerical treatment of Equations (4), (6), (11) and (12). Due to the intricacy of this approach, the subsection is further subdivided for the sake of clarity. The numerical details of the individual steps, notably the respective discrete differential operators employed, of the ISPH algorithm are presented in Section 4.3.1. Aspects to be considered for the numerical solution of the pressure Poisson equation (PPE), which plays a pivotal role in the projection-based ISPH method, are covered in Section 4.3.2.

### 4.3.1. Discrete ISPH Scheme

While the substrate is locally molten as a result of the thermalisation of the interference irradiation provided by the laser pulse, the melt flow is calculated using the ISPH algorithm explained below. The particle positions and velocities are evolved for the completely fluid particles inside the molten pool, whereas the dynamic pressure values are also determined for the surrounding edge particles. In accordance with the sole solution of the energy Equation (17) presented in Section 4.2, the transition from the old time step $t^{*,n}$ to the new time step $t^{*,n+1} = t^{*,n} + \Delta t^*$ is considered for the numerical treatment of Equations (17)–(20) shown here.

At first, the fluid particles are advected to intermediate positions based on the old velocity

$$\overline{x}_i^* = x_i^{*,n} + v_i^{*,n} \Delta t^*. \tag{67}$$

Considering the acceleration due to viscous and body forces, an intermediate velocity field

$$\overline{v}_i^* = v_i^{*,n} + \left( Pr \boldsymbol{L}_i^* \left( 1, v_j^{*,n} \right) + PrRa \frac{T_i^{*,n} - T_1^*}{1 - T_1^*} \boldsymbol{e}_z \right) \Delta t^* \tag{68}$$

is predicted, which is not divergence-free. Subsequently, the PPE for enforcing zero velocity divergence

$$L_i^* \left( 1, p_{\mathrm{dyn},j}^{*,n+1} \right) = \frac{1}{\Delta t^*} D_i^{-*} \left( \overline{v}_j^* \right) \tag{69}$$

is solved at the fluid particles inside and the edge particles surrounding the molten pool. The intermediate velocity is then corrected using the gradient of the determined dynamic pressure field to obtain a divergence-free velocity field

$$v_i^{*,n+1} = \overline{v}_i^* - \Delta t^* \boldsymbol{G}_i^{+*} \left( p_{\mathrm{dyn},j}^{*,n+1} \right). \tag{70}$$

After that, the particle positions are updated using both old and new velocity fields

$$x_i^{*,n+1} = x_i^{*,n} + \left( v_i^{*,n} + v_i^{*,n+1} \right) \frac{\Delta t^*}{2}.$$ (71)

The specific enthalpy at the new time step is calculated

$$h_i^{*,n+1} = h_i^{*,n} + \Delta t^* \left( L_i^* \left( 1, T_j^{*,n+1} \right) + \dot{Q}_i^{*,n+1} \right)$$ (72)

as discussed above in Section 4.2. To avoid a too orderly particle motion along the streamlines, the particle positions are slightly shifted [69]

$$\widetilde{x}_i^{*,n+1} = x_i^{*,n+1} + C\alpha^* R_i, \quad R_i = \sum_{j=1}^{N_i} \left( \frac{r_i^{*,\mathrm{av}}}{r_{ij}^{*,n+1}} \right)^2 n_{ij}, \quad r_i^{*,\mathrm{av}} = \frac{1}{N_i} \sum_{j=1}^{N_i} r_{ij}^{*,n+1}, \quad n_{ij} = \frac{x_{ij}^{*,n+1}}{r_{ij}^{*,n+1}},$$ (73)

with a constant $C \in [0.01, 0.1]$, the shifting magnitude $\alpha^* = \max_j \left| v_j^{*,n+1} \right| \Delta t^*$ and the shifting vector $R_i$ depending on the average particle spacing $r_i^{*,\mathrm{av}}$ and the unit vectors $n_{ij}$. A truncated Taylor series expansion is then employed to adjust the velocity values to the final particle positions (The discrete gradient operator in Equation (74) is characterised by $G_i^{-*} \left( v_j^{*,n+1} \right) = \left[ G_i^{-*} \left( v_{x,j}^{*,n+1} \right) \ G_i^{-*} \left( v_{z,j}^{*,n+1} \right) \right]^{\mathsf{T}}$.)

$$\widetilde{v}_i^{*,n+1} = v_i^{*,n+1} + G_i^{-*} \left( v_j^{*,n+1} \right) \left( \widetilde{x}_i^{*,n+1} - x_i^{*,n+1} \right).$$ (74)

In addition, an analogous adjustment is performed for the specific enthalpy values

$$\widetilde{h}_i^{*,n+1} = h_i^{*,n+1} + \left( G_i^{-*} \left( h_j^{*,n+1} \right) \right)^{\mathsf{T}} \left( \widetilde{x}_i^{*,n+1} - x_i^{*,n+1} \right).$$ (75)

Finally, the adjusted specific enthalpy values are considered for the determination of the molten and vaporised mass fractions and the particle temperatures $T_i^{*,n+1}$ according to Equations (26)–(28).

It is observed that the procedure described above uses two different sets of positions for the particle approximations during each time step. The discrete differential operators employed in the velocity prediction, solution of the PPE and velocity correction steps in Equations (68)–(70) rely on the advected particle positions given in Equation (67). On the other hand, the discrete summations in the specific enthalpy update, position correction, velocity and specific enthalpy adjustment steps in Equations (72)–(75) are based on the updated particle positions calculated in Equation (71).

In addition, after the specific enthalpy and temperature update in Equation (72), the temperature gradient is determined using the symmetric gradient operator $G_i^{-*} \left( T_j^{*,n+1} \right)$, the horizontal component being required for the Marangoni boundary condition given by Equation (21) in the next time step.

Furthermore, the ISPH scheme exposed above is subject to restrictions on the dimensionless time step size. Rewriting Equation (59) in dimensionless form, the conditions to be respected are given as

$$\Delta t^* = \min \left( 0.25 \frac{l_{\mathrm{sm}}^*}{v_{\mathrm{max}}^*}, 0.25 \min_i \sqrt{\frac{l_{\mathrm{sm}}^*}{|f_i^*|}}, 0.125 \min_i \frac{l_{\mathrm{sm}}^{*\,2}}{Pr}, 0.125 \min_i l_{\mathrm{sm}}^{*\,2}, 0.25 \frac{Oh}{Pr} \sqrt{\frac{l_{\mathrm{sm}}^{*\,3}}{2\pi}} \right),$$ (76)

where the Ohnesorge number $Oh = \nu / \sqrt{\gamma L / \rho}$ emerges in the dimensionless surface tension condition.

### 4.3.2. Solution of PPE

Applying the dimensionless form of the discrete Laplacian and symmetric divergence operator in Equations (47) and (48), respectively, the zero velocity divergence PPE (69) to be solved is written out as

$$
2 \sum_{j=1}^{N_i} \frac{m_j^*}{\rho_j^*} \frac{p_{\mathrm{dyn},i}^{*,n+1} - p_{\mathrm{dyn},j}^{*,n+1}}{\left| \overline{x}_i^* - \overline{x}_j^* \right|} \frac{\partial W \left( \overline{x}_i^* - \overline{x}_j^*, l_{\mathrm{sm}}^* \right)}{\partial \left| \overline{x}_i^* - \overline{x}_j^* \right|} = \frac{1}{\Delta t^*} \sum_{j=1}^{N_i} \frac{m_j^*}{\rho_j^*} \frac{\left( \overline{v}_j^* - \overline{v}_i^* \right) \cdot \left( \overline{x}_i^* - \overline{x}_j^* \right)}{\left| \overline{x}_i^* - \overline{x}_j^* \right|} \frac{\partial W \left( \overline{x}_i^* - \overline{x}_j^*, l_{\mathrm{sm}}^* \right)}{\partial \left| \overline{x}_i^* - \overline{x}_j^* \right|}. \tag{77}
$$

The presence of Equation (77), which all fluid particles inside and edge particles bounding the molten pool are to satisfy, constitutes a linear system of equations $A p_{\mathrm{dyn}}^{*,n+1} = b$ for the dynamic pressure values.

It should be evident from the explanations given below that the concept of a system matrix $A$, although it is not assembled in the numerical code, is meaningful for the solution of the PPE. The dynamic pressure field to be determined by solving Equation (77) is subject to a homogeneous Neumann boundary condition, the dimensionless form of the first condition in Equation (14), at the molten pool edges, as mentioned in Section 2.1. For this type of boundary condition, the technique adopted in this work requires that the dynamic pressure values at the edge particles are assigned to the respective dummy particles situated beyond the molten pool edges in the outward normal direction.

The idea is to write the linear system consisting of an equation of the type given in Equation (77) for each fluid and edge particle $i$ in the form $A p_{\mathrm{dyn}}^{*,n+1} = b$ with a square matrix $A$, where $p_{\mathrm{dyn}}^{*,n+1}$ is the solution vector comprising the dynamic pressures of all fluid and edge particles $i$. Contemplating on the left hand side of Equation (77) for fluid and edge particles, the structure of the system matrix can be understood from the following observations. To begin with, as the derivative of the kernel function vanishes at the origin, the self-interaction of a fluid or edge particle $i$ is neglected.

The notation $\{i_{\mathrm{D}}\}$ is introduced for the set of dummy particles associated with an edge particle $i$. In particular, the interactions of an edge particle $i$ with the dummy particles $j \in \{i_{\mathrm{D}}\}$ are disregarded as the dynamic pressure values cancel out. Apart from the interaction with a (different) edge particle $j$, each contribution due to the interaction of a fluid or edge particle $i$ with any dummy particle $j' \in \{j_{\mathrm{D}}\}$ by means of the value $p_{\mathrm{dyn},j}^{*,n+1}$ is allocated to the associated edge particle $j$ with equal dynamic pressure. These considerations lead to the non-zero entries of the matrix $A$ listed in Table 1. In detail, the off-diagonal elements $a_{ij}$ in the second row of Table 1 are valid for both fluid and edge particles $i$.

**Table 1.** Non-zero elements in matrix $A$ of linear equation system corresponding to the pressure Poisson equation (PPE).

| Particle | Role of Particle | |
| :---: | :---: | :---: |
| | **Fluid Particle** | **Edge Particle** |
| $i$ | $a_{ii} = 2 \sum_{\substack{j=1 \\ j \neq i}}^{N_i} \frac{m_j^*}{\rho_j^*} \frac{1}{\overline{r}_{ij}^*} \frac{\partial W \left( \overline{r}_{ij}^*, l_{\mathrm{sm}}^* \right)}{\partial \overline{r}_{ij}^*}$ | $a_{ii} = 2 \sum_{\substack{j=1, j \neq i \\ j \notin \{i_{\mathrm{D}}\}}}^{N_i} \frac{m_j^*}{\rho_j^*} \frac{1}{\overline{r}_{ij}^*} \frac{\partial W \left( \overline{r}_{ij}^*, l_{\mathrm{sm}}^* \right)}{\partial \overline{r}_{ij}^*}$ |
| $j$ | $a_{ij} = -2 \frac{m_j^*}{\rho_j^*} \frac{1}{\overline{r}_{ij}^*} \frac{\partial W \left( \overline{r}_{ij}^*, l_{\mathrm{sm}}^* \right)}{\partial \overline{r}_{ij}^*}$ | $a_{ij} = -2 \frac{m_j^*}{\rho_j^*} \frac{1}{\overline{r}_{ij}^*} \frac{\partial W \left( \overline{r}_{ij}^*, l_{\mathrm{sm}}^* \right)}{\partial \overline{r}_{ij}^*} - 2 \sum_{j' \in \{j_{\mathrm{D}}\}} \frac{m_{j'}^*}{\rho_{j'}^*} \frac{1}{\overline{r}_{ij'}^*} \frac{\partial W \left( \overline{r}_{ij'}^*, l_{\mathrm{sm}}^* \right)}{\partial \overline{r}_{ij'}^*}$ |

The following properties of the system matrix $A$ can be inferred from its elements presented in Table 1 and further reflexions. In general, the matrix is comparatively sparse since the number of interactions $N_i$ of neighbouring particles with particle $i$ is much less than the total number of particles representing the molten pool and its edges. Nevertheless, the situation that each fluid particle can in principle interact

with any other fluid or edge particle throughout the simulation leads to the fact that the matrix is not banded. It is evident from Table 1 that the main diagonal entries $a_{ii}$ of the matrix $A$ are negative, whereas its off-diagonal elements $a_{ij}$ are positive.

The matrix $A$ is weakly diagonally dominant, as $\forall i : |a_{ii}| = \sum_{j \neq i} |a_{ij}|$. More precisely, the sum of the elements in each row of $A$ is zero. Therefore, the vector $x_0 = (1, 1, \ldots, 1)^\mathsf{T}$ is in the kernel of $A$. Consequently, the matrix $A$ is not of full rank, i.e., $A$ is singular, according to the rank-nullity theorem. This problematic feature of $A$ arises for confined flows in the absence of a Dirichlet boundary condition on the pressure [47,79].

Now consider a fluid particle $i$ situated sufficiently close to an edge so that there is at least one dummy particle $j'$ beyond edge particle $j$ interacting with particle $i$, i.e., $\exists j' \in \{j_D\} : W\left(\bar{r}_{ij'}^*, l_{\text{sm}}^*\right) > 0$, then $a_{ij} = -2 \frac{m_j^*}{\rho_j^*} \frac{1}{\bar{r}_{ij}^*} \frac{\partial W\left(\bar{r}_{ij}^*, l_{\text{sm}}^*\right)}{\partial \bar{r}_{ij}^*} - 2 \sum_{j' \in \{j_D\}} \frac{m_{j'}^*}{\rho_{j'}^*} \frac{1}{\bar{r}_{ij'}^*} \frac{\partial W\left(\bar{r}_{ij'}^*, l_{\text{sm}}^*\right)}{\partial \bar{r}_{ij'}^*}$. On the contrary, there is no additional interaction allocated to the one with fluid particle $i$ in the row giving rise to Equation (77) for edge particle $j$, i.e., $a_{ji} = -2 \frac{m_i^*}{\rho_i^*} \frac{1}{\bar{r}_{ji}^*} \frac{\partial W\left(\bar{r}_{ji}^*, l_{\text{sm}}^*\right)}{\partial \bar{r}_{ji}^*}$. Consequently, as $a_{ij} \neq a_{ji}$, the matrix $A$ is non-symmetric.

As the system matrix $A$ is not regular as mentioned above, it should be regularised to solve the linear system of equations. This is performed through a slight reinforcement of the diagonal entries and by subtracting from the right hand side of the equation system the mean of its entries [47].

In this work, the linear system with a resulting non-symmetric and regular system matrix is solved iteratively using the BiCGStab algorithm [80]. However, a numerical instability of BiCGStab denoted as pivot breakdown [81] is often encountered during the solution of the PPE. For this reason, a stabilisation of the algorithm is implemented, which detects a pivot near-breakdown situation and consequently restarts the BiCGStab iteration.

## 5. Simulation Results

In the following, the model presented above is applied to investigate DLIP of metallic substrates using a single nanosecond pulse. For the sake of clarity, the present section is divided into two subsections. First, the parameters of the DLIP process are indicated in Section 5.1 along with the material properties considered for both stainless steel and aluminium substrates as well as the actual values of the dimensionless numbers. Subsequently, the details of the numerical investigation and the results of DLIP simulations by means of (I)SPH are presented in Section 5.2.

### 5.1. Model Parameters

As already indicated above, the interference of two coherent laser beams giving rise to a sinusoidal intensity distribution is studied in this work. The considered parameters of the DLIP process, particularly with regard to the laser heat source, are given in Table 2. Note that the periodicity $\Lambda$ is given as a function of the laser wavelength $\lambda$ and the angle of intersection $\theta$ between the interfering beams according to Equation (2). Due to the specification of the thermal diffusion length as the characteristic length scale, the Fourier number defined in Equation (22) and given in Table 2 depends only on the pulse duration and the considered physical time, irrespective of the substrate.

**Table 2.** Process parameters of single-pulse direct laser interference patterning (DLIP) of metallic substrate.

| Process Parameter | Symbol | Value |
|---|---|---|
| wavelength | $\lambda$ | 355 nm |
| full angle between beams | $\theta$ | 0.071 rad |
| periodicity of interference pattern | $\Lambda$ | 5 µm |
| energy density (fluence) per beam | $\Phi_0$ | 0.3 J/cm$^2$ |
| pulse duration (FWHM) | $\tau_p$ | 10 ns |
| pulse time | $t_p$ | 50 ns |
| simulation duration | $t_{end}$ | 200 ns |
| initial substrate temperature | $T_0$ | 298.15 K |
| gravitational acceleration | $g$ | 9.81 m/s$^2$ |
| Fourier number | $Fo$ | 5.0 |

Concerning the considered substrates, AISI 304 stainless steel and high-purity aluminium, the relevant material properties are listed in Table 3. If available, the respective temperature-dependent values are averaged over the entire interval from the room temperature to the vapourisation point to obtain the reference density $\rho_{ref}$, specific heat $c_{p,ref}$, thermal conductivity $\kappa_{ref}$ and thermal diffusivity $a_{ref}$. Based on the latter, the thermal diffusion length $L$ is calculated and, along with process parameters and material properties from Tables 2 and 3, employed to determine the dimensionless quantities stated in Table 4.

Furthermore, in case of the stainless steel substrate, the absence of surfactants, e.g., sulphur, in the alloy is assumed and a constant temperature coefficient of surface tension d$\gamma$ /d$T$ is considered, see Table 3. The nonlinear temperature dependency of both surface tension and its temperature coefficient [82] in the presence of surface-active elements as well as the effect of this additional complexity on DLIP results of stainless steel substrates are explored in a separate investigation.

**Table 3.** Material properties of stainless steel and aluminium substrates.

| Material Property | Symbol | AISI 304 Stainless Steel | High-Purity Aluminium | Unit | Refs. |
|---|---|---|---|---|---|
| solidus temperature | $T_s$ | 1673 | $T_m$: 933.35 | K | [83] |
| liquidus temperature | $T_l$ | 1727 | | K | [83] |
| vapourisation temperature | $T_v$ | 3273 | 2792 | K | [84,85] |
| enthalpy of fusion | $L_f$ | 251 | 397 | kJ/kg | [83,84] |
| enthalpy of vapourisation | $L_v$ | 6500 | 10860 | kJ/kg | [84,86] |
| density | $\rho_{ref}$ | 7262 | 2228 | kg/m$^3$ | [83,87,88] |
| specific heat | $c_{p,ref}$ | 704 | 1077 | J/(kgK) | [83,89] |
| thermal conductivity | $\kappa_{ref}$ | 26.8 | 139.5 | W/(mK) | [83,90] |
| thermal diffusivity | $a_{ref}$ | $5.24 \cdot 10^{-6}$ | $5.75 \cdot 10^{-5}$ | m$^2$/s | |
| dynamic viscosity (at $T_l$ or $T_m$) | $\eta$ | $7.0 \cdot 10^{-3}$ | $1.38 \cdot 10^{-3}$ | Pas | [83,91] |
| kinematic viscosity (at $T_l$ or $T_m$) | $\nu$ | $1.02 \cdot 10^{-6}$ | $5.80 \cdot 10^{-7}$ | m$^2$/s | |
| volumetric thermal expansion coefficient | $\beta$ | $8.5 \cdot 10^{-5}$ | $9.8 \cdot 10^{-5}$ | 1/K | [92,93] |
| temperature coefficient of surface tension | d$\gamma$ /d$T$ | $-4.3 \cdot 10^{-4}$ | $-2.75 \cdot 10^{-4}$ | N/(mK) | [82,94] |
| absorption coefficient (at 355 nm) | $\alpha$ | $9.56 \cdot 10^{7}$ | $1.52 \cdot 10^{8}$ | 1/m | [85] |
| reflectivity (at 355 nm) | $R$ | 0.556 | 0.925 (at $T_0$) | – | [85,95] |

**Table 4.** Thermal diffusion length and dimensionless quantities of DLIP model for metallic substrates.

| Quantity | Symbol | AISI 304 Stainless Steel | High-Purity Aluminium |
|---|---|---|---|
| thermal diffusion length | $L$ | 457.8 nm | 1.517 µm |
| Laser number | $La$ | 37.7151 | 152.657 |
| solid-liquid phase change number | $Ph_{s/l}$ | 0.119849 | 0.147849 |
| liquid-vapour phase change number | $Ph_{l/v}$ | 3.10367 | 4.04443 |
| Prandtl number | $Pr$ | 0.1947 | 0.0101 |
| Rayleigh number | $Ra$ | $2.31448 \cdot 10^{-8}$ | $1.86969 \cdot 10^{-7}$ |
| Marangoni number | $Ma$ | 8.29744 | 9.76488 |

In particular, the absorptivity of aluminium exhibits a very distinct temperature dependency. Although the reflectivity of aluminium is 92.5% at the wavelength $\lambda = 355$ nm at room temperature [85,95] as indicated in Table 3, it decreases to 65% at the melting point [84]. Therefore, the effect of temperature on the reflectivity of aluminium is considered by employing the following relation [12]

$$R = \begin{cases} 1.048 - 3.989 \cdot 10^{-4} T \ / \text{K} & T_0 \le T < T_{\text{m}}, \\ 0.65 & T_{\text{m}} \le T. \end{cases} \tag{78}$$

**Table 5.** Range of the processing parameter fluence investigated for metallic substrates, complemented by specification of corresponding Laser number and finely discretised near-surface zone depth.

| Substrate | Quantity | Value | | | | | | | Unit |
|---|---|---|---|---|---|---|---|---|---|
| | $2\Phi_0$ | 0.4 | 0.5 | 0.6 | 0.7 | 0.8 | 0.9 | 1.0 | J/cm$^2$ |
| AISI 304 | $La$ | 25.1434 | 31.4292 | 37.7151 | 44.0009 | 50.2868 | 56.5726 | 62.8585 | – |
| | fine zone | 270 | 270 | 320 | 320 | 320 | 370 | 370 | nm |
| | $2\Phi_0$ | 0.6 | 0.7 | 0.8 | 0.9 | 1.0 | 1.1 | 1.2 | J/cm$^2$ |
| Al 99.99% | $La$ | 152.657 | 178.100 | 203.543 | 228.986 | 254.429 | 279.871 | 305.314 | – |
| | fine zone | 0.77 | 0.77 | 1.02 | 1.02 | 1.27 | 1.27 | 1.52 | µm |

*5.2. Numerical Investigation of DLIP*

Simulations of DLIP experiments using a single nanosecond pulse are performed by means of the approach presented above. The fluence of the interference pattern to be specified allows local melting of the substrate at the interference maxima without significant vapourisation of material. To this end, the suitable range of fluences is identified from the literature [96,97] and this processing parameter is reported in Table 5. In addition, the Laser number and the depth of the near-surface zone, which is discretised by the finest particles initially arranged on an equidistant Cartesian lattice, provided for the molten pool computation, are indicated in Table 5 for the different fluences and metallic substrates.

A few comments on the principal characteristics of the DLIP simulations are given beforehand. Due to the absorption and thermalisation of the incident radiation, the substrate is rapidly heated after the onset of the laser pulse and locally melted. The surface temperature at the interference maximum is saved in each time step of the SPH simulation to understand its evolution. In the case of an elevated laser fluence and a high absorptivity, the maximum surface temperature may even reach the vapourisation point. As opposed to the previous work [35], the vapourisation of particles by their respective deactivation is disregarded in the present investigation, although the simulations suggest that the latent heat of vapourisation is

completely inserted into surface particles for elevated energy densities starting from 0.6 J/cm$^2$ for stainless steel and from 1.0 J/cm$^2$ for aluminium.

In the presence of molten metal, the nonuniform surface temperature gives rise to surface tension variations, the highest values being found at the boundaries of the molten pool in accordance with the negative temperature coefficient of surface tension. The surface tension gradients cause shear stresses at the surface inducing an outward flow from the centre of the melt pool, i.e., at the interference maximum, to its edge and, owing to continuity, a backward flow at its bottom. To get an idea of the importance of melt pool convection, the melt pool dimensions and the extremes of the horizontal particle velocity found below the surface particles are recorded in each time step of the SPH simulation while metal melt is present. After the action of the laser pulse, the temperature field inside the substrate tends to homogenise as a result of heat conduction. This phenomenon in particular entails the resolidification of the molten material, starting from the melt pool edges.

### 5.2.1. Computational Results for Steel Substrate

Considering the stainless steel substrate, SPH simulations of single pulse DLIP are performed for laser energy densities ranging from 0.4 J/cm$^2$ to 1.0 J/cm$^2$. The predictions of the time-dependent surface temperature at the interference maximum are presented in Figure 3 for this range of fluences along with the temporal intensity variation of the laser pulse. Except for the lowest fluence, the surface is heated to the vapourisation temperature at the interference maximum as indicated in the figure.

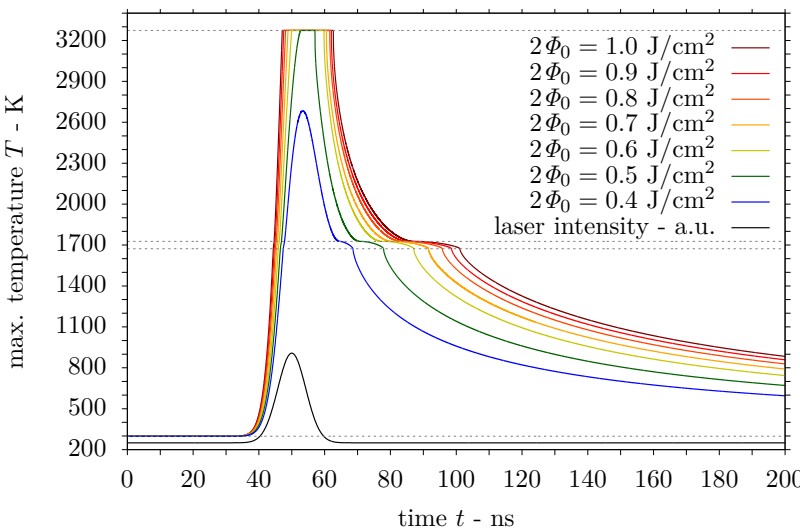

**Figure 3.** Temporal evolution of surface temperature at interference maximum during single pulse DLIP of AISI 304 stainless steel substrate as predicted by the SPH model for different laser fluences. The variation of the time-dependent laser pulse intensity is indicated for the sake of clarity.

The melt pool width and depth obtained from the SPH computations are presented in Figure 4a,b, respectively, for DLIP of stainless steel in the considered range of energy densities. For plotting the trends in Figure 4, each discrete value of a melt pool dimension is employed only once at an intermediate point in time to avoid a stair-step appearance of the graphs, particularly in Figure 4b. It is observed from Figure 4 that the presence of the melt pool is considerably prolonged for an enhanced laser fluence. In addition, the melt pool width and even more its depth increase distinctly with the energy density for low and moderate fluences up to 0.7 J/cm$^2$. Nevertheless, the melt pool developing due to the effect of the laser pulse is rather shallow with an aspect ratio (depth/width) of the order of 0.1.

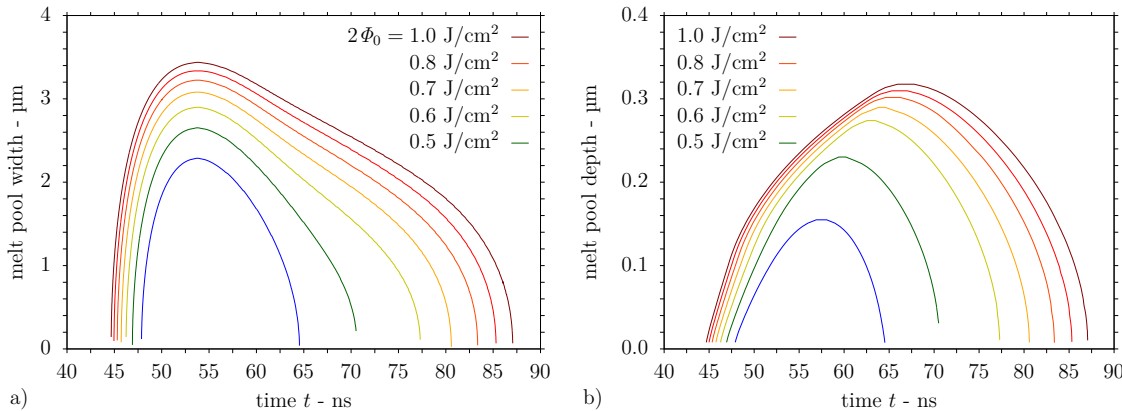

**Figure 4.** Evolution of melt pool dimensions during single pulse DLIP of AISI 304 stainless steel as determined by the SPH model for different laser fluences: (**a**) melt pool width and (**b**) melt pool depth.

The resulting maximum (outward) horizontal velocity trends during melt pool convection are reported in Figure 5 for the investigated laser fluences. The velocity variations in Figure 5 show a superlinear growth of the maximum horizontal velocity just below the surface for an increased laser energy density. In particular, the velocity trends exhibit an excessive increase of the maximum horizontal velocity magnitude starting at a time later than 55 ns, i.e., after the laser pulse duration (FWHM), except for the simulation considering the lowest fluence.

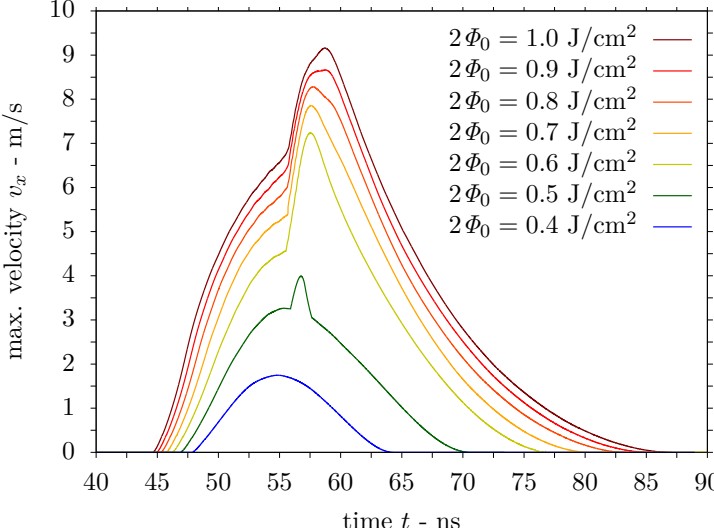

**Figure 5.** Maximum horizontal velocity magnitude below the melt pool surface as determined by the SPH model of DLIP considering AISI 304 stainless steel substrate and different laser fluences.

The latter observation is examined in detail for a fluence of 0.5 J/cm$^2$ in Figure 6, where the evolution of the temperature, the horizontal temperature gradient and the velocity at the melt pool surface is presented in the relevant temporal range. After the onset of melt cooling, temperature gradients as high as several 1000 K/μm, see Figure 6b, arise at particles in the close proximity of a central superficial zone at the vapourisation point, see Figure 6a. By means of the Marangoni boundary condition in Equation (15), these excessive temperature gradients lead to a pair of inner horizontal velocity extrema reaching a magnitude of 4 m/s, in addition to the pre-existing pair of outer horizontal velocity extrema with absolute values not much higher than 3.5 m/s, see Figure 6c. As the inner velocity maximum temporarily exceeds the

outer one, it is the former that is reported in Figure 5 during this time interval. However, for the moderate fluence considered, the inner velocity extrema eventually decay in the absence of a surface zone at the vapourisation point and the adjacent high temperature gradients.

In addition, to provide some more detail to the horizontal velocity trends shown in Figure 5, the temporal evolution of the magnitude of inner and outer horizontal velocity maxima is presented in Figure 7 for different laser fluences. For a moderate fluence of 0.5 J/cm$^2$, Figure 7a shows that the outer and inner extrema coexist for more than 4 ns, which is also observed in Figure 6c. However, the magnitude of the inner velocity extrema exceeds the one at the outer extrema only for 1.7 ns according to Figure 7a, which is evident from Figure 5 as well. On the other hand, the horizontal velocity magnitudes presented in Figure 7b reveal that the inner extrema emerge within fractions of a nanosecond and soon dominate for elevated laser energy densities. In this situation, the (outer) horizontal velocity extrema existing from the onset of molten pool convection subsequently vanish, cf. Figure 7b.

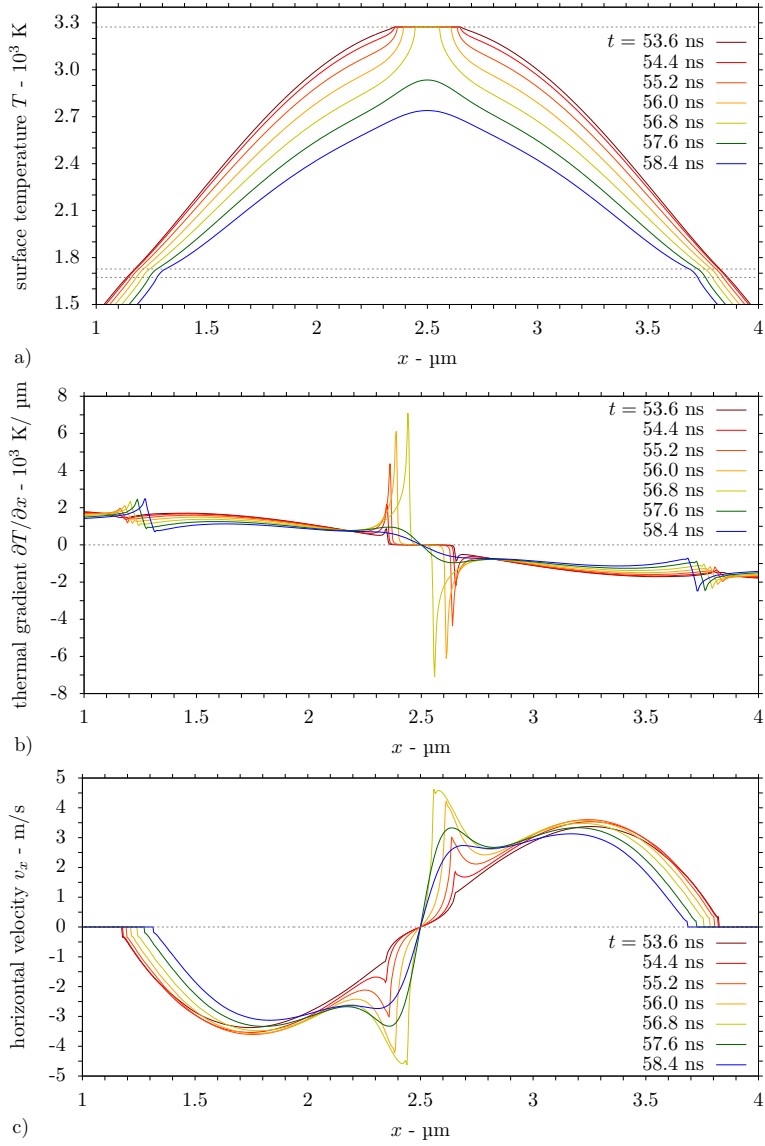

**Figure 6.** Evolution of (**a**) temperature, (**b**) horizontal temperature gradient, and (**c**) velocity at the melt pool surface during DLIP of AISI 304 stainless steel substrate for the laser fluence $2\Phi_0 = 0.5$ J/cm$^2$.

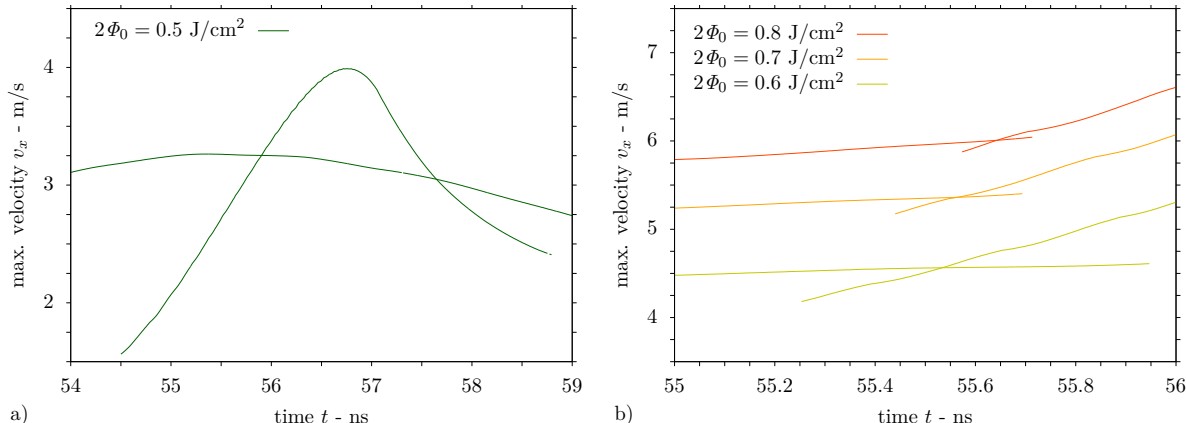

a)                                                                                     b)

**Figure 7.** Occurrence of second extremum of horizontal velocity below the surface closer to the centre of the melt pool during DLIP of AISI 304 stainless steel substrate: (**a**) Coexistence of outer (permanent) and inner extrema for moderate fluence. (**b**) Onset and persistence of inner extrema for higher fluences.

### 5.2.2. Computational Results for Aluminium Substrate

The results of DLIP simulations performed by SPH for the high-purity aluminium substrate and energy densities ranging from 0.6 J/cm$^2$ to 1.2 J/cm$^2$ are outlined in the following. At first, Figure 8 presents the temporal evolution of the maximum surface temperature for different fluences. Moderate temperatures of the aluminium surface at the interference maximum are reported in Figure 8, even at elevated fluences. This result can be attributed to the low absorptivity, particularly in the solid state, the high thermal diffusivity and the enhanced specific heat and latent heat of fusion of aluminium. The trends in Figure 8 further indicate a rapid homogenisation of the substrate temperature after the laser pulse action, the maximum surface temperature returning to the melting point comparatively early.

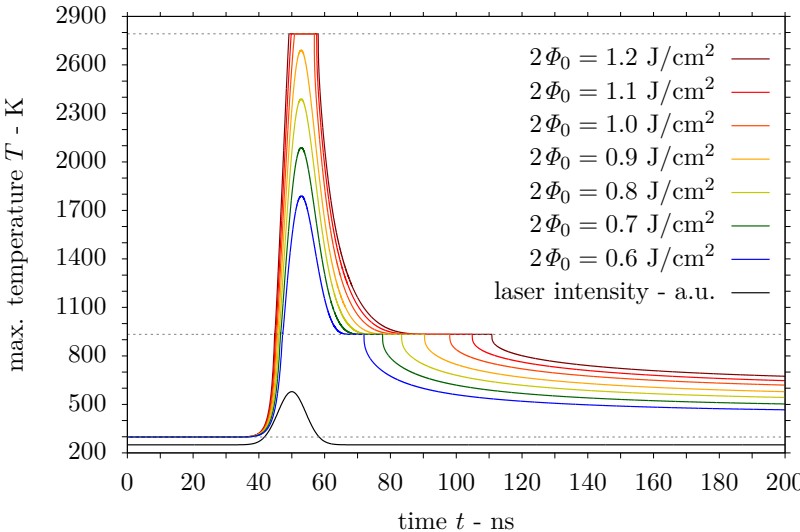

**Figure 8.** Surface temperature at interference maximum during single pulse DLIP of high-purity aluminium substrate as a function of time according to SPH predictions for different laser fluences.

The melt pool dimensions, i.e., its width and depth, predicted by the SPH simulations are plotted against time in Figure 9a,b, respectively. For the graphs in Figure 9, the individual discrete values of the melt pool dimensions are again considered only once at a central point in time to evade a stair-step

presentation. The figure indicates that an increase in the laser fluence results in an extended presence as well as larger dimensions of the melt pool. Concerning DLIP of aluminium, the data presented in Figure 9b reveal that the melt pool is comparatively deep, where an aspect ratio of the melt pool in the range of 0.2 to 0.3 can be deduced from Figure 9.

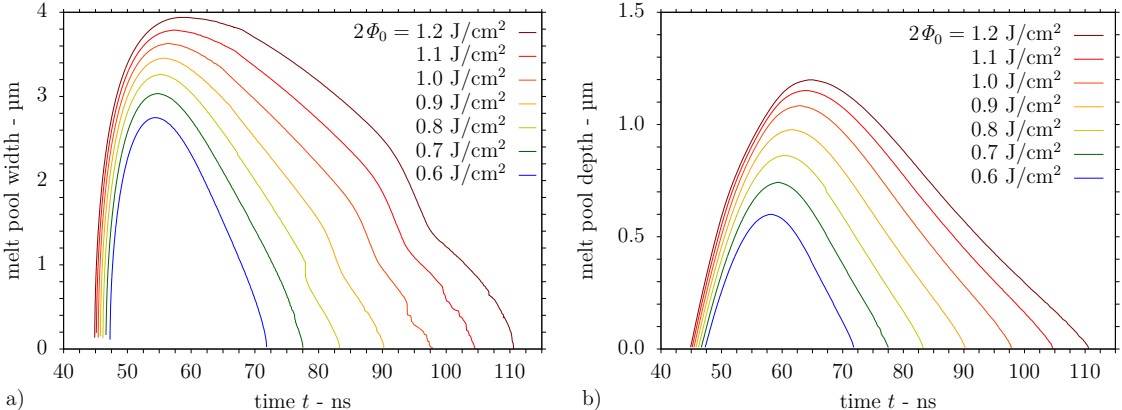

**Figure 9.** Evolution of melt pool dimensions during single pulse DLIP of high-purity aluminium as determined by the SPH model for different laser fluences: (**a**) melt pool width and (**b**) melt pool depth.

Furthermore, the maximum horizontal velocity magnitudes at particles below the surface are presented in Figure 10 for the considered laser fluences. The velocity trends displayed in Figure 10 confirm a superlinear growth of the horizontal velocity below the surface for an increased laser energy density. In fact, remarkable velocity magnitudes beyond 10 m/s are attained even at moderate laser fluences employed for DLIP of aluminium substrates. This observation is explored in greater detail in Figure 11, where the evolution of the temperature, the horizontal temperature gradient and the velocity are presented along the melt pool surface and the adjacent solid material.

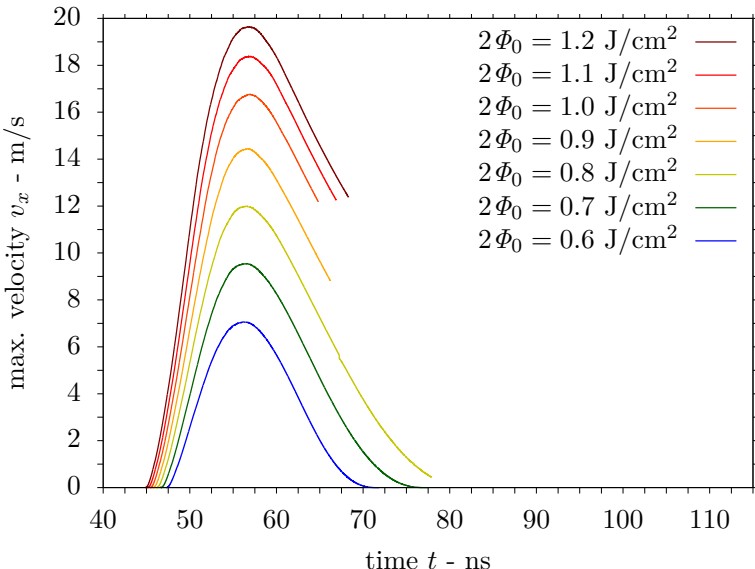

**Figure 10.** Maximum horizontal velocity magnitude below the melt pool surface as computed by SPH during DLIP of high-purity aluminium substrate for different laser fluences.

More precisely, the aforementioned quantities are displayed in Figure 11 for DLIP of aluminium using an energy density of 1.0 J/cm² in the temporal range characterised by the onset of cooling and the passing of the maximum melt pool width and horizontal velocity, cf. Figures 9a and 10. At the beginning of the temperature field homogenisation, Figure 11b shows that high temperature gradients arise in the close vicinity of a central surface zone at the vapourisation point, cf. Figure 11a. However, these thermal gradients do not result in the complete formation of separate horizontal velocity extrema adjacent to the central high temperature region, see Figure 11c. Apart from or in the absence of this surface region, Figure 11b exposes a consistent temperature gradient on both sides of the interference maximum. Owing to the temperature dependence of surface tension, this steady temperature gradient along the melt pool surface gives rise to a smoothly varying horizontal velocity profile with its extrema apart from the centre and closer to the edges of the melt pool, see Figure 11c.

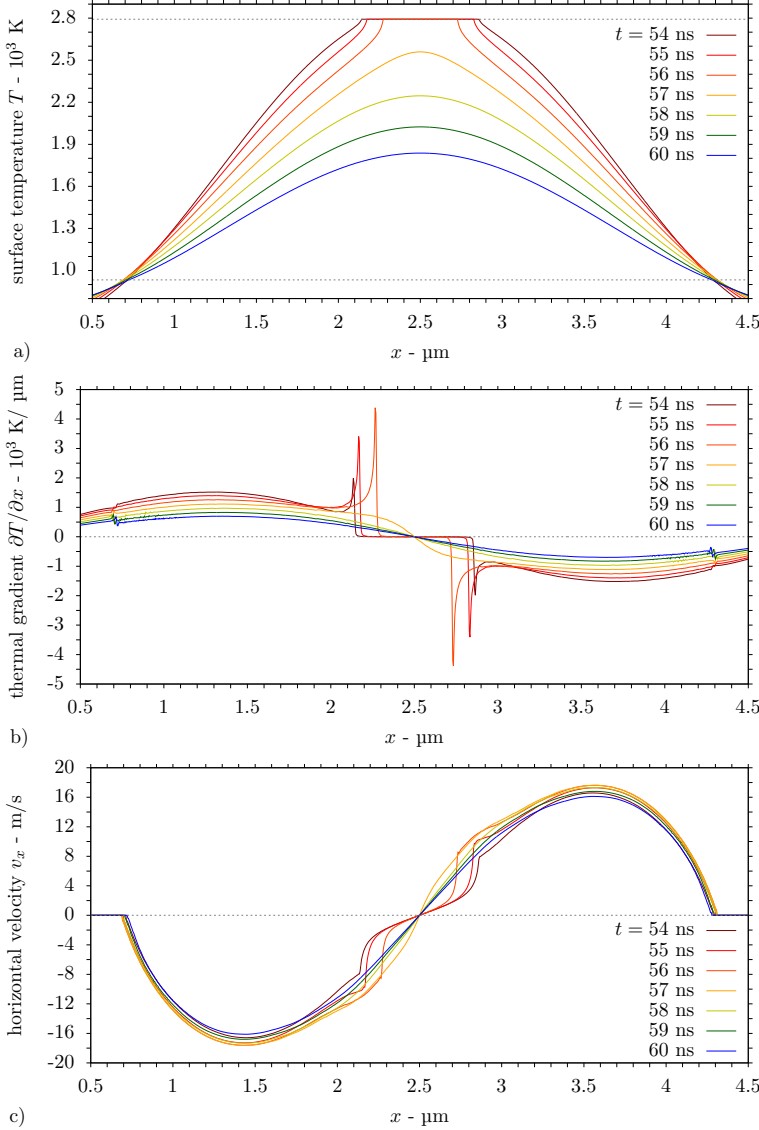

**Figure 11.** Evolution of (**a**) temperature, (**b**) horizontal temperature gradient and (**c**) velocity at the melt pool surface during DLIP of high-purity aluminium substrate for the laser fluence $2\Phi_0 = 1.0$ J/cm².

## 6. Discussion

The results presented above suggest that stainless steel and aluminium represent two very dissimilar substrate materials with distinct behaviour when subject to laser interference irradiation. To begin with, the maximum surface temperature exhibits a more pronounced sensitivity to the laser fluence for the aluminium substrate, see Figure 8, as compared to the results in Figure 3 for stainless steel. This observation may be attributed to the high reflectivity of aluminium and the self-enhancing effect of its temperature-dependent absorptivity during laser heating. In addition, the thermal diffusivity of aluminium, being one order of magnitude higher than the one of stainless steel, facilitates conductive heat transfer, which leads to moderate maximum surface temperatures. On the other hand, the high thermal diffusivity of aluminium results in an enlarged melt pool, see Figure 9, particularly in a larger melt pool depth, when compared with the melt pool dimensions predicted for stainless steel in Figure 4.

Furthermore, the maximum velocity magnitude trends determined for the aluminium substrate, see Figure 10, indicate a more distinct sensitivity to the laser fluence in comparison with the results for stainless steel presented in Figure 5. The maximum horizontal velocity magnitudes presented in Figure 10 for the aluminium melt, which notably exceed 10 m/s for elevated fluences, are at least twice as high as the ones in Figure 5 predicted for stainless steel. Contemplating on the Marangoni boundary condition in Equation (15), this difference is explained by the considerably smaller dynamic viscosity of liquid aluminium, which is only one fifth of the value for stainless steel according to Table 3. The higher velocity values for the liquid aluminium are even bounded by the lower magnitudes of the temperature coefficient of surface tension, cf. Table 3, and the horizontal temperature gradient, see Figure 11c. In conjunction with the markedly deeper melt pool, the increased horizontal melt velocities suggest that aluminium is a delicate substrate material, which requires a careful selection of the fluence for DLIP. Actually, the high melt velocities predicted in this work provide an explanation for the irregular surface morphology observed for aluminium substrates, see Figure 12c [96], unlike the nearly perfect repetitive topography of stainless steel, see Figure 12a [35], after DLIP at moderate fluences.

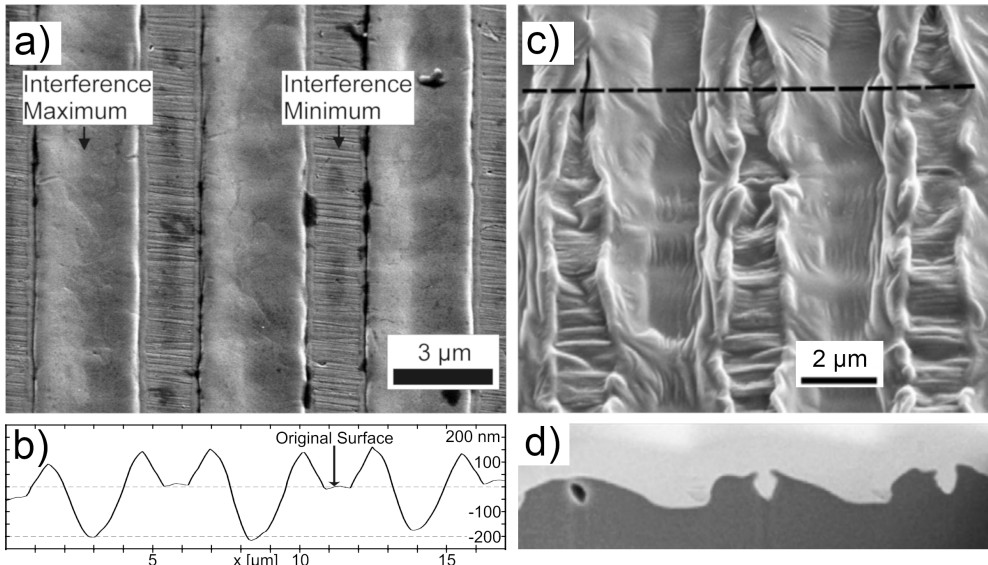

**Figure 12.** Surface morphologies after two-beam DLIP with a single laser pulse, $\lambda = 355$ nm, $\tau_\mathrm{P} = 10$ ns: (**a**,**b**) on AISI 304 stainless steel substrate with a period of 5.05 µm and laser fluence 0.6 J/cm$^2$ [35], (**c**,**d**) on high-purity aluminium substrate with a period of 4.5 µm and laser fluence 1.015 J/cm$^2$ [96], scanning electron micrographs of (**a**,**c**) substrate surfaces, (**d**) cross section at dashed line in (**c**) prepared by focussed ion beam and detail of (**b**) surface profile measured by confocal laser scanning microscope.

The surface microstructure and profile reprinted in Figure 12a,b were observed after a single pulse DLIP experiment with a periodicity of $\Lambda = 5.05$ μm and a fluence of 0.6 J/cm$^2$ on stainless steel. The width of the molten zone at the steel surface presented in Figure 12a is approximately 3.2 μm [35], which is in reasonable agreement with the maximum melt pool width of 2.9 μm, see Figure 4a, in the simulation. The surface profile in Figure 12b shows that melt is displaced from the interference maxima towards the interference minima [35]. The height difference between the original surface and the resulting depression is approximately 200 nm [35] and of the same order as the melt pool depth of 270 nm, see Figure 4b, computed in this work.

Concerning the high-purity aluminium substrate, the surface morphology and profile shown in Figure 12c,d, respectively, were obtained using a slightly smaller periodicity of $\Lambda = 4.5$ μm and a fluence of 1.015 J/cm$^2$. For this reason, only the height of the resulting surface structures is compared with the melt pool depth predicted for aluminium. A structure height of approximately 0.8 μm can be inferred from the cross section in Figure 12d, which suggests that the volume of metal melt is considerably larger in the case of aluminium. The melt pool depth of 1.1 μm predicted in the present simulations for a fluence of 1.0 J/cm$^2$, see Figure 9b, is in agreement with the measured depth of the surface microstructure on aluminium after DLIP with a periodicity of $\Lambda = 4.5$ μm, as reported in [96].

The present results are in reasonable agreement with previous work on DLIP of metals [12,96,97]. In detail, the maximum surface temperatures of aluminium during DLIP are in accordance with values computed using the FEM model, which were presented in [97]. However, lower maximum surface temperatures were predicted in [97] for stainless steel, probably owing to a considerably higher thermal diffusivity value employed. The magnitude of the horizontal temperature gradient at the surface of the stainless steel and aluminium substrates is in line with the thermal gradient determined in [97]. Further, the melt pool depth during DLIP of stainless steel presented in Figure 4b is of the same order of magnitude as the molten depth calculated in [12], in spite of the fact that the latter results were obtained for a smaller periodicity.

On the other hand, the molten depths determined from the thermal simulations for an aluminium substrate in [12] are notably lower than the present results in Figure 9b, although a comparable periodicity was considered. This deviation may be attributed to a higher thermal diffusivity of aluminium employed in [12], which entails a more homogeneous temperature field with less pronounced maxima and, therefore, a reduced absorption of the incident radiation. Nevertheless, the present depths of the aluminium melt pool are in the same size range as the structure depths measured after DLIP experiments on aluminium substrates using a similar periodicity, which were reported in [96].

Furthermore, the validity of the predicted velocity magnitudes is confirmed in the following. Considering the molten material as a shear layer, a characteristic horizontal velocity of the thermocapillary motion is obtained from the Marangoni boundary condition in Equation (15) as [98,99]

$$u_{\text{tc}} = \frac{\delta_{\text{m}}}{\eta} \frac{\mathrm{d}\gamma}{\mathrm{d}T} \frac{\partial T}{\partial x}, \tag{79}$$

where $\delta_{\text{m}}$ denotes a height related to the melt pool. If the height $\delta_{\text{m}}$ is specified as half of the molten film thickness, $u_{\text{tc}}$ in Equation (79) represents an average horizontal velocity of melt displacement [12,98]. For the present simulations, excessive horizontal velocity values would result if the melt pool depth was employed in Equation (79). Instead, it is suggested to substitute the thickness of the velocity boundary

layer near the surface for $\delta_m$. In addition, a time scale for thermocapillary convection is given by the ratio of half the interference pattern periodicity and the characteristic velocity magnitude [12,100]

$$t_{tc} = \frac{\Lambda/2}{|u_{tc}|} = \frac{\eta \Lambda}{2\delta_m \left| \frac{d\gamma}{dT} \frac{\partial T}{\partial x} \right|}, \tag{80}$$

where the horizontal temperature gradient is often assessed by relating a temperature difference to a width, e.g., $\partial T/\partial x \approx \Delta T /\Lambda/2$ [12], the resulting expression being proportional to $\Lambda^2/4$ [100].

Considering the simulations presented in detail in Section 5, the horizontal velocity magnitudes at the time $t = 57$ ns are estimated by Equation (79) using the material properties from Table 3. For stainless steel subject to the laser fluence $2\Phi_0 = 0.5$ J/cm$^2$, the average temperature gradient $\partial T/\partial x \approx -1240$ K/µm and $\delta_m \approx 68$ nm result in $u_{tc} \approx 5.13$ m/s. In case of the high-purity aluminium substrate and the energy density $2\Phi_0 = 1.0$ J/cm$^2$, the average temperature gradient $\partial T/\partial x \approx -900$ K/µm and $\delta_m \approx 115$ nm yield $u_{tc} \approx 20.6$ m/s. The maximum surface velocity magnitudes computed in the ISPH simulations, as may be identified from Figures 6c and 11c, are 32% and 15%, respectively, lower than these values. These deviations may be attributed to the transient character of the surface temperature distribution and the short duration of melt pool convection. Employing $\Lambda/2 = 2.5$ µm and the aforementioned velocities in Equation (80), the time scale for thermocapillary flow equals $t_{tc} \approx 487$ ns for stainless steel and $t_{tc} \approx 121$ ns for aluminium.

## 7. Conclusions

The present work demonstrates the use of SPH in the simulation of heat transfer and fluid flow during nanosecond pulsed DLIP of metallic substrates on the basis of a dimensionless formulation of the governing equations. For single-pulse treatment, the simulation results reveal a distinct behaviour of the dissimilar substrates stainless steel and high-purity aluminium. Particularly in the case of processing aluminium, the predictions confirm that thermocapillary convection is characterised by substantial velocity magnitudes beyond 10 m/s even at moderate laser fluences. Consequently, this outward flow from the centre of the melt pool surface, at the interference maximum, towards its edges is a conceivable structuring mechanism effective during DLIP of aluminium using a nanosecond pulse at low and moderate energy densities.

On the contrary, the higher absorptivity and lower thermal conductivity of stainless steel lead to high surface temperatures up to the vapourisation point near the interference maximum, even at moderate laser fluences. In addition, thermocapillary convection is less pronounced in liquid steel owing to its high dynamic viscosity. Therefore, it is more difficult to distinguish between the effects of the thermocapillary flow and the recoil pressure induced by vapourisation on the melt displacement during DLIP of stainless steel at moderate energy densities. The further investigation of the roles of thermocapillary convection and vapourisation-induced recoil pressure in the structuring of metal surfaces by DLIP ultimately necessitates the modelling of the melt pool surface deformation.

The numerical results presented in this work are compared with surface microstructures obtained by material characterisation after DLIP experiments on stainless steel and high-purity aluminium. In agreement with the observed surface morphologies, the simulations indicate a significantly deeper molten bath and a more effective melt displacement mechanism for the aluminium substrate. The predicted velocity magnitudes, which suggest a notable outward flow due to thermocapillary forces at the surface of the aluminium melt pool, are confirmed by a theoretical estimation.

In this work, an ISPH approach is deliberately employed to simulate the melt flow during DLIP of metals. This choice is motivated by virtue of the meaningful, physical pressure field computed using the ISPH technique. It is expected that an accurate pressure field is essential for the further study of melt displacement during DLIP, particularly for modelling the effect of the recoil pressure. The present model

could serve as a basis for the prospective investigation. Although the SPH method is inherently capable of describing the evolution of a free surface in fluid flow, the application of a projection-based ISPH approach involves fundamental difficulties in this context. Provided that these issues can be eliminated, the method presented herein can be appropriately extended to simulate the melt displacement during DLIP of metals as well.

**Author Contributions:** The research was conceptualised by A.F.L. and C.D., the methodology and software were developed by C.D., the results were discussed by A.F.L. and C.D., the original draft was written by C.D., the manuscript was reviewed and edited by A.F.L. and C.D. All authors have read and agreed to the published version of the manuscript.

**Funding:** This research received no external funding. The APC was borne in the frame of Open Access Funding by the Publication Fund of the TU Dresden and the Saxon State and University Library Dresden (SLUB).

**Acknowledgments:** The authors thank Achim Mahrle (TU Dresden, Institute of Manufacturing Technology) for the original suggestion of applying SPH to the modelling of laser material processing. Further, the first author acknowledges Theresa Jähnig (TU Dresden, Institute of Manufacturing Technology), Hartmut Krause, Subhashis Ray and Eric Werzner (TU Bergakademie Freiberg, Institute of Thermal Engineering) for the helpful discussions.

**Conflicts of Interest:** The authors declare no conflicts of interest.

## Abbreviations

The following abbreviations are used in this manuscript:

| | |
|---|---|
| AISI | American Iron and Steel Institute |
| BiCGStab | biconjugate gradient stabilised |
| CG | conjugate gradient |
| DLIP | direct laser interference patterning |
| FEM | finite element method |
| FWHM | full width at half maximum |
| ISPH | incompressible smoothed particle hydrodynamics |
| PPE | pressure Poisson equation |
| SPH | smoothed particle hydrodynamics |
| WCSPH | weakly compressible smoothed particle hydrodynamics |

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
