# Peer review of "An Incompressible Smoothed Particle Hydrodynamics (ISPH) Model of Direct Laser Interference Patterning"

_computation, doi:10.3390/computation8010009_

Round 1

Reviewer 1 Report

Authors are solving heat and mass transfer equations with ISPH, for heating small scale metalic material. They have mentioned that the physics/nature behind this phenomenon is not quite clear, which is true. And hence, we don't quite for sure know which equations we have to solve OR how to solve them and get the correct data. Then, my discussion follows the classical fact that why/how we should trust your solution while you did not even for a tiny bit compare it with experimental data? If that is not possible, then these sophisticated solutions and graphs (which are great) are just pure mathematics without validation (and may be a type of art). Hence, they may not have solid physical/practical value. If authors cannot provide a type of validation, using available experimental data (or at least numerical solutions) I must reject the paper.

Author Response

Dear Reviewer,

Please see the responses to the comments in the attachment.

Reviewer 2 Report

The manuscript is well written, with a clear message delivered to the reader. I judge the adopted methods appropriate. In my opinion the manuscript however warrants publication after major revision as next specified:

Abstract

Briefly give assumptions of the Marangoni boundary condition. Give more emphasis to your conclusions consequently on the basis of the obtained results.

Introduction

The cited literature is comprehensive but somewhere not up-to-date; I would suggest to take also into account the following research papers:

D Violeau: Fluid Mechanics and the SPH Method - Theory and Applications, 616 pages, Oxford University Press, 2012, ISBN: 10: 0199655529

It is true that [16] – [18] are good reviews of SPH, nevertheless I would add Violeau’s book (even is substitution of [17]) as it is a good one.

LINE 62. “… involving fluid flow, as in turbomachinery, coastal and hydraulic engineering […].”

It is true that SPH has been applied in fluid dynamics during the last couple of decades. I would comprehend here

Marongiu, J.-C., Leboeuf, F., Caro, J., Parkinson, E. 2010. Free surface flows simulations in Pelton turbines using a hybrid SPH-ALE method. Journal of Hydraulic Research 48, 40-49.

Pugliese Carratelli, E., Viccione, G., Bovolin, V. 2016. Free surface flow impact on a vertical wall: a numerical assessment. Theoretical and Computational Fluid Dynamics 30(5), 403-414.

The literature review on SPH applied to DLIP is excellent, comprehending meaningful papers by Rogers and Souto-Iglesias.

Smoothed Particle Hydrodynamics

The “Fundamentals of the SPH method” is well written. However, I do not totally agree on condition (38) as the kernel could need to be smoother (that is of C^2 class as in the case of Stable anisotropic heat conduction in smoothed particle hydrodynamics, by Sergei Biriukov and Daniel J. Price, 2018)

I would underline the fact that one of the main advantages of SPH is to have the derivative of the kernel function on the RHS of eq. (44) which is a known function together with its derivatives as well explained by eqs. (41).

LINE 237. “However, this procedure gives rise to considerable noise in the pressure field, since small fluctuations in the density field result in large pressure fluctuations due to the stiffness of the equation of state [55].”

This is true in the classical form of WCSPH. As matter of facts, from 2008 on (the year of publication [55]) a number of papers have been proposed to overcome such a drawback see for instance:

Colagrossi, A., Bouscasse, B., Antuono, M., Marrone, S. 2012. Particle packing algorithm for SPH schemes. Computer Physics Communications 183(8), 1641-1653

Aristodemo, F., Meringolo, D.D., Groenenboom, P., Lo Schiavo, A., Veltri, P., Veltri, M. 2015. Assessment of dynamic pressures at vertical and perforated breakwaters through diffusive SPH schemes. Mathematical Problems in Engineering 2015,305028

Sun, P.N., Colagrossi, A., Marrone, S., Antuono, M., Zhang, A.-M.             2019. A consistent approach to particle shifting in the δ-Plus-SPH model. Computer Methods in Applied Mechanics and Engineering 348, 912-934

Nothing is said about the neighbourhood definition, which is essential in SPH computations, in terms of running costs, see for instance:

Viccione, G., Bovolin, V., Pugliese Carratelli, E. 2008. Defining and optimizing algorithms for neighbouring particle identification in SPH fluid simulations. International Journal for Numerical Methods in Fluids 58(6), 625-638

Domínguez, J.M., Crespo, A.J.C., Gómez-Gesteira, M., Marongiu, J.C. 2011. Neighbour lists in smoothed particle hydrodynamics. International Journal for Numerical Methods in Fluids 67(12), 2026-2042

Numerical solution of governing equations

LINE 322. “The vertical distance between these two particles can be written as”

Please explain the reason why the vertical distance is here assumed.

Simulation results

Results are clearly presented and the following discussion is well structured. However, in my opinion there are two mission points: first of all, results need to be “grid independent” meaning in the present case that they would not depend (or let’s say, they change so little) on the assumed interparticle distance. Second key point: any validation with experimental data? This aspect would add major value to the manuscript.

Conclusions

Could be extended on the basis of a broaden discussion, see previous point.

Author Response

(The authors gave the same response as above.)

Round 2

Reviewer 1 Report

Good job.

Reviewer 2 Report

Dear Authors, the manuscript has been significantly improved. I acknowledge the efforts made.

Regards